# A Model of Place Field Reorganization During Reward Maximization

## Abstract

When rodents learn to navigate in a novel environment, a high density of place fields emerges at reward locations, fields elongate against the trajectory, and individual fields change spatial selectivity while demonstrating stable behavior. Why place fields demonstrate these characteristic phenomena during learning remains elusive. We develop a normative framework using a reward maximization objective, whereby the temporal difference (TD) error drives place field reorganization to improve policy learning. Place fields are modeled using Gaussian radial basis functions to represent states in an environment, and directly synapse to an actor-critic for policy learning. Each field's amplitude, center, and width, as well as downstream weights, are updated online at each time step to maximize cumulative reward. We demonstrate that this framework unifies the three disparate phenomena observed in navigation experiments. Furthermore, we show that these place field phenomena improve policy convergence when learning to navigate to a single target and relearning multiple new targets. To conclude, we develop a normative model that recapitulates several aspects of hippocampal place field learning dynamics and unifies mechanisms to offer testable predictions for future experiments.

## 1 Introduction

A place field is canonically described as a localized region in an environment where the firing rate of a hippocampal neuron is maximal and robust across trials (O'Keefe, 1978; O'Keefe & Dostrovsky, 1971). Classically, each neuron has a unique spatial receptive field such that the population activity can describe an animal's allocentric position within the environment (Moser et al., 2015). Ablation studies demonstrate that the hippocampal representation is useful for learning to navigate to new targets (Morris et al., 1982; Packard & McGaugh, 1996; Steele & Morris, 1999). Importantly, each field's spatial selectivity evolves with experience in a new environment before stabilizing in the later stages of learning (Frank et al., 2004). Specifically, a high density of place fields emerge at reward locations (Gauthier & Tank, 2018; Lee et al., 2020; Sosa et al., 2023), place fields elongate backward against the trajectory (Mehta et al., 1997; Priestley et al., 2022), and individual field's spatial selectivity continues to change or "drift" even when animals demonstrate stable behavior (Geva et al., 2023; Kentros et al., 2004; Krishnan & Sheffield, 2023; Mankin et al., 2012; Ziv et al., 2013). Although disparate mechanisms have been proposed to model these phenomena, a framework that can unify their phenomena and clarify their computational role remains elusive.

Here, we propose a normative model for spatial representation learning in hippocampal CA1, given its role in representing salient spatial information (Dong et al., 2021; Dupret et al., 2010). Our primary contributions are as follows:

- We develop a two-layered reinforcement learning model to study spatial representation learning by place fields (Fig.1A). The first layer contains a population of Gaussian radial basis functions that transform continuous spatial information into a relevant representational substrate, which feed into the actor-critic network in the second layer that uses these representations to maximize cumulative discounted reward. Besides the actor and critic weights, each place field's firing rate, center of mass and width is optimized by the temporal difference error.

- Our model recapitulates three experimentally-observed neural phenomena during task learning: the emergence of high place field density at rewards, elongation of fields against the trajectory, and drifting fields that do not affect task performance.

- We analyze the factors that influence these representational changes: a low number of fields drives greater spatial representation learning, each place field's firing rate reflects the value of that location, and increasing noise magnitude during field parameter updates causes a monotonic decrease in population vector correlation but non-monotonic change in behavior.

- We demonstrate that optimizing place field widths and amplitudes enhances reward maximization and policy convergence. However, field parameter optimization alone is insufficient for learning to navigate to new targets. Introducing noisy field parameter updates improves new target learning, suggesting a functional role for noise.

## 2 RELATED WORKS

**Anatomically constrained architecture for navigation.** Learning to navigate involves the hippocampus encoding spatial information and its strong glutamatergic connections to the striatum (Floresco et al., 2001; Lisman & Grace, 2005). The ventral and dorsal regions of the striatum are associated with value estimation and stimulus-response associations, functioning similarly to a critic and an actor, respectively (Houk et al., 1994; Joel et al., 2002; Niv, 2009). Additionally, dopamine neurons in the Ventral Tegmental Area influence plasticity in the striatal synapses (Reynolds et al., 2001; Russo & Nestler, 2013). This anatomical insight has led to the design of a biologically plausible navigation model, where place fields connect directly to an actor-critic framework, and synapses are modulated by the TD error (Arleo & Gerstner, 2000; Brown & Sharp, 1995; Foster et al., 2000; Frémaux et al., 2013; Kumar et al., 2022). Furthermore, recent evidence shows direct dopaminergic projections to the hippocampus to modulate place cell activity, strengthening the case for navigation models with adaptive place fields (Kempadoo et al., 2016; Krishnan et al., 2022; Palacios-Filardo & Mellor, 2019; Sayegh et al., 2024). How upstream information from the entorhinal cortex influences place field representations for policy learning needs clarity (Bush et al., 2015; Fiete et al., 2008).

**Field density increases near reward locations.** As animals learn to navigate in a 1D track, a high density of place fields emerge at reward locations. We define density to be both the number of fields (Gauthier & Tank, 2018; Sosa et al., 2023) and the peak firing rate of each field (Lee et al., 2020). Reward location based reorganization was observed in hippocampal CA1 and not in CA3 (Dupret et al., 2010).

**Fields learn to encode future occupancy.** As animals traverse a 1D track towards a reward, most CA1 fields increase in size and their center of mass shift backwards against the trajectory of motion (Frank et al., 2004; Mehta et al., 1997; Priestley et al., 2022). A proposal for this behavior is that fields initially encoding only location $x_t$ are learning to also encode the previous location $x_{t-1}$, and hence are coding future location occupancy $p(x_{t+1}|x_t)$ (Mehta et al., 2000; Stachenfeld et al., 2017). While algorithms such as the successor representation (Dayan, 1993) learn to predict the transition structure (Gardner et al., 2018; Gershman, 2018), the representation is dependent on a predefined navigation policy. Hence, a complete normative argument—including policy learning—for why fields exhibit this behavior is still lacking.

**Fields drift during stable behavior.** After animals reach a certain performance criterion in navigating to a reward location, the spatial selectivity of individual place fields changes across days, even though animals exhibit stable behavior (de Snoo et al., 2023; Geva et al., 2023; Kentros et al., 2004; Mankin et al., 2012; Ziv et al., 2013). A proposal is that these fields continue to drift within a degenerate solution space while the overall representational manifold or the chosen performance metric remains stable (Kappel et al., 2015; Masset et al., 2022; Pashakhanloo & Koulakov, 2023; Qin et al., 2023; Rokni et al., 2007). However, a model that demonstrates stable navigation learning behavior with drifting fields is absent. Furthermore, why drifting fields might be useful is still unexplored.

**Place fields versus place cells.** Several experiments have shown that place fields along the dorsoventral axis have different widths (Jung et al., 1994) and are also involved in navigation (Contreras et al., 2018; Harland et al., 2021), while newer experiments challenge the canonical definition that a place cell only has one place field (Eliav et al., 2021). As a simple starting point, in this work we study spatial representational learning using Gaussian place fields, instead of place cells.

## 3 TASK AND MODEL SETUP

Most navigational experiments involve an animal moving from a start location to a target location to receive a reward, either in a one-dimensional (1D) track or a two-dimensional (2D) arena. Similarly, our agents receive their true position at every time step ($t$) described by the variable (scalar $x_t$ in 1D, vector $\boldsymbol{x}_t$ in 2D), and have to learn a policy ($\pi$) that specifies the actions to take ($g_t$) to move from a start location (e.g. $x_{start} = -0.75$, Fig. 1A green dash) to a target with reward values following a Gaussian distribution ($x_r = 0.5, \sigma_r = 0.05$, Fig. 1A red area). The agent outputs a discrete one-hot vector $g_t$ (left or right in 1D and left, right, up or down in 2D), which causes its motion to be discrete, similar to a trajectory in a grid world. To model smooth trajectories in a continuous space as an animal's behavior, we use a low-pass filter to smooth $g_t$ using a constant $\alpha_{env} = 0.2$ after scaling for maximum displacement using $v_{max} = 0.1$:

$$x_{t+1} = x_t + a_t \quad , \quad a_{t+1} = (1 - \alpha_{env})a_t + \alpha_{env}v_{max}g_t \,, \tag{1}$$

similar to past works (Foster et al., 2000; Frémaux et al., 2013; Kumar et al., 2022; 2024b; Zannone et al., 2018). To track an agent's reward maximization performance during navigational learning we compute the true cumulative discounted reward ($G = \sum_{t=0}^{T} \gamma^t r_{t+1}$) for each trial using $\gamma = 0.9$ as the discount factor, which is similar to tracking the cumulative reward. The trial is terminated if the trial time reaches a threshold $T_{max}$ or when the total reward achieved $\sum_{t=0}^{T} r_{t+1}$ reaches a threshold $R_{max}$. For further details, see App. A.

### 3.1 PLACE FIELDS AS SPATIAL FEATURES

The agent represents space through $N$ place fields, which have spatial selectivity modeled as simple Gaussian bumps and tile the environment:

$$\phi_i(x_t) = \alpha_i^2 \exp(-||x_t - \lambda_i||_2^2/2\sigma_i^2) \,, \tag{2}$$

where $\alpha$, $\lambda$ and $\sigma$ set the amplitude, center, and width respectively. Two types of place field distributions were initialized: (1) Homogeneous population of constant values for amplitudes $\alpha_i = 0.5$, widths $\sigma_i = 0.1$, and centers uniformly tiling the environment $\boldsymbol{\lambda} = [-1, ..., 1]$ (Foster et al., 2000; Frémaux et al., 2013; Kumar et al., 2022; 2024b; Zannone et al., 2018). (2) Heterogeneous population with amplitudes, widths and centers drawn from uniform random distributions between $[0, 1]$, $[10^{-5}, 0.1]$, $[-1, 1]$ respectively. These ranges are consistent with experimental data where place fields were 20 cm to 50 cm wide in a small environment (Frank et al., 2004; Lee et al., 2020; Mehta et al., 1997; Sosa et al., 2023). 2D place fields have scalar amplitudes, two dimensional vectors for center, and square covariance matrices for the width (Menache et al., 2005). Refer to App. A.

### 3.2 POLICY LEARNING USING AN ACTOR-CRITIC

To model an animal's trial-and-error based learning behavior, we adopt the reinforcement learning framework, specifically the actor-critic (Arleo & Gerstner, 2000; Brown & Sharp, 1995; Foster et al., 2000; Frémaux et al., 2013; Kumar et al., 2022; 2024b). The critic linearly weights place field activity using a vector $w_i^v$ to estimate the value of the current location

$$v(x_t) = \sum_i^N w_i^v \phi_i(x_t) \,. \tag{3}$$

The value of a location corresponds to the expected cumulative discounted reward for that location. The actor has $M$ units, each specifying a movement direction. In the 1D and 2D environments, $M = 2$ and $M = 4$ respectively to code for opposing directions in each dimension e.g. left versus right and up versus down. Each actor unit $a_j$ linearly weights the place field activity such that the matrix $W_{ji}^\pi$ computes the preference for moving in the $j$-th direction

$$a_j(x_t) = \sum_i^N W_{ji}^\pi \phi_i(x_t) \quad , \quad P_j = \frac{\exp(a_j)}{\sum_k^M \exp(a_k)} \,, \tag{4}$$

with the probability of taking an action computed using a softmax. A one-hot vector $g_j$ is sampled from the action probability distribution $P$ as in Foster et al. (2000), making this policy stochastic. $w_i^v$ and $W_{ji}^\pi$ were initialized by sampling from a random normal distribution $\mathcal{N}(0, 10^{-5})$.

### 3.3 BIOLOGICALLY RELEVANT REWARD MAXIMIZATION LEARNING OBJECTIVE

The objective of our agent is to maximize the expected cumulative discounted reward $\mathcal{J}^G = \mathbb{E}[G_t] = \mathbb{E}[\sum_{k=0}^{T} \gamma^k r_{t+1+k}]$. To achieve this goal in an online manner, our agent uses the stan-

dard actor-critic algorithm using the expected temporal difference objective (refer to App. A):

$$\mathcal{J}^{TD} = \mathbb{E}\left[\sum_t^T r_{t+1} + \gamma v(x_{t+1}) - v(x_t)\right] = \mathbb{E}[\sum_t^T \delta_t]. \tag{5}$$

which reduces variance and speeds up policy convergence (Dayan & Abbott, 2005; Mnih et al., 2016; Schulman et al., 2017; Sutton & Barto, 2018; Wang et al., 2018). The TD error is also biologically relevant, as the responses of midbrain dopamine neurons resemble TD reward prediction error (Amo et al., 2022; Gershman & Uchida, 2019; Montague et al., 1996; Schultz et al., 1997; Starkweather & Uchida, 2021). The actor learns a reward maximizing policy by ascending the gradient of the policy log likelihood, modulated by the TD error. To accurately estimate the TD error and critique policy learning, the critic learns a value function by minimizing the squared TD error $\mathcal{L} = \mathbb{E}\left[\sum_t^T \frac{1}{2}\delta_t^2\right]$.

As our agent uses a single population of place fields, these fields must learn spatial features that enhance both policy and value learning. The field parameters $\theta = \{\alpha, \lambda, \sigma\}$ and the policy weights $W^\pi$, $w^v$ are updated by gradient ascent using a joint objective modified from Wang et al. (2018):

$$\nabla_{\theta, W^\pi, w^v} \mathcal{J} = \nabla_{\theta, W^\pi} \mathcal{J}^{TD} - \nabla_{\theta, w^v} \mathcal{L} = \mathbb{E}\left[\sum_t^T \left(\nabla_{\theta, W^\pi} \log \pi(g_t|x_t) + \nabla_{\theta, w^v} v(x_t)\right) \cdot \delta_t\right], \tag{6}$$

with $\nabla_{w^v} \mathcal{J}^{TD} = 0$ and $\nabla_{W^\pi} \mathcal{L} = 0$. We estimate all parameter gradients online, and provide the explicit update equations for each parameter in App. A. The learning rates for the actor-critic and place field parameters can be the same (Sup. Fig. 13). For theoretical analysis, we assume a separation of timescales between learning the actor-critic weights and updating place field parameters (App. B). This approach stabilizes place field representation learning, and is consistent with Dong et al. (2021)'s observation that rodent behavior converges faster than place field representations.

## 4 RESULTS

### 4.1 A HIGH DENSITY OF FIELDS EMERGES NEAR THE REWARD LOCATION

We first examine the neural phenomenon where a high density of place fields emerges at the reward location. Field density is defined by the distribution of field centers of mass (COM) (Gauthier & Tank, 2018), which we estimate using Gaussian kernel smoothing. Figure 1B shows how our agent's track occupancy, field density, mean firing rate, and individual field's spatial selectivity change when learning to navigate in a 1D track from the start location $x_{start} = -0.75$ to the target at $x_r = 0.5$, when only optimizing place field centers ($\Delta\lambda$). In the early stages of learning, the agent spends a higher proportion of time at the start location with sporadic exploration towards the reward. Despite this behavior, a high field density and mean firing rate rapidly emerges at the target from a homogeneous field population within the first few trials. Individual fields at the reward location shift closer to the target (Fig. 1F), as seen in (Gauthier & Tank, 2018; Sosa et al., 2023), in contrast to fields at non-rewarded locations. As learning progresses and the agent spends a higher proportion of time at the reward location, field density and mean firing rate at the start location also begins to rise slightly, although it remains lower than at the reward location, replicating the two-peaked field distribution in (Gauthier & Tank, 2018). A high density at the reward location followed by the start location robustly emerges in heterogeneous place field populations when all the field parameters ($\Delta\lambda, \Delta\alpha, \Delta\sigma$) are optimized (Fig. 1B right, Sup. Fig. 2B). Similar field dynamics are observed in a 2D arena with an obstacle where agents have to navigate to a target from a starting location (Fig. 1C). When optimizing all the field parameters in a homogeneous population, a high field density rapidly emerges at the reward location to increase goal representation as seen in (Dupret et al., 2010), followed by gradual reorganization of field density along the agent's trajectory back to the start location.

Interestingly, increasing the number of fields in a heterogeneous place field population reduced the average density (Fig. 1D, Sup. Fig. 1) and mean firing rate (Sup. Fig. 4D) that emerges near the reward location ($R > 0.01$). This is because as the number of fields increase, the agent goes into a weak feature learning regime (Sup. Fig. 4) in which feature learning does not contribute to additional advantage. While experiments can record thousands of place fields, only a small fraction of fields, between 80 to 150, show reward-relative reorganization (Gauthier & Tank, 2018; Lee et al., 2020; Sosa et al., 2023). Conversely, the density and mean firing rate are proportional to the reward magnitude (blue versus green), and inversely proportional to the reward location width (red versus purple) as a narrower target might require higher discriminability for the agent to maximize rewards.

To understand why place fields exhibit these dynamics, we perform a perturbative approximation to the place field parameter changes under TD learning updates (Bordelon et al., 2024; Menache et al.,

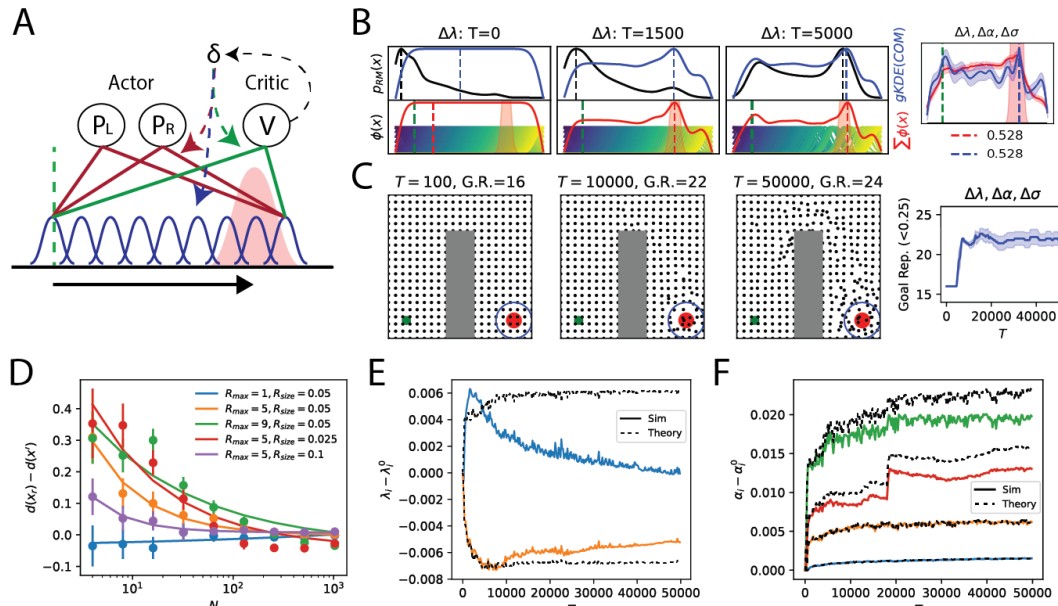

Figure 1: **Emergence of high field density at the reward location with learning.** (**A**) The task is to navigate from the start (green dash) to the target (red area) to receive rewards whose magnitude follows a Gaussian distribution. The agent contains $N$ Gaussian place fields (blue) which synapse to an actor (red) and critic (green) to learn the policy and value function respectively. The temporal difference error $\delta$ is used to update parameters. (**B-C**) Example of an increase in place field density at the reward location during learning in a (**B**) 1D track (Gauthier & Tank, 2018; Lee et al., 2020), and (**C**) 2D arena (Dupret et al., 2010) with an obstacle (gray). (**B**) When optimizing field centers (Top row) In the early learning phase ($T = 100$), the agent spends a high proportion of time ($p_{RM}(x)$, black) at the start location with a constant field density (gKDE(COM), blue) throughout the track. As learning proceeds ($T = 2000, 50000$), a higher field density emerges at the reward and start location when only optimizing field centers ($\Delta\lambda$). (Bottom row) Evolution of individual field centers and mean firing rates ($\sum \phi(x)$, red). (Right) A high field density and mean firing rate emerges at the reward location, followed by the start location, for a heterogeneous place field population when all field parameters are optimized ($\Delta\lambda, \Delta\alpha, \Delta\sigma$). (**C**) The density similarly evolves in the 2D arena when all field parameters are optimized. In the early learning phase ($T < 10000$), centers of mass (COM, black dots) shift to the target, causing a high density to emerge at the reward (right). In the later learning phase, the rest of the COM align along the trajectory. The start and reward locations and radius for goal representation (G.R.) are marked by green, red and blue circles in the leftmost plot. (**D**) As the number of fields increases, the average field density ($d(x) = gKDE(COM)$) near the reward location $x_r$ compared to non-reward location $x'$ decreases for the heterogeneous population. The density decreases when the reward magnitude decreases ($R_{max} = 1, 5, 9$: blue, orange, green) and reward location's size increases ($R_{size} = 0.025, 0.05, 0.1$: red, orange, purple). (**E-F**) Example of field dynamics when an agent ($N = 512$) navigates a 1D track. (**E**) Fields initialized before ($\lambda_i = 0.5$, blue) and after ($\lambda_i = 0.6$, orange) the target move forward and backward respectively, increasing the density near the target. (**F**) Fields closest to the reward ($\lambda_i = 0.5, 0.6$: green and red) show a rapid and high amplification compared to other fields ($\lambda_i = -0.75, 0.0$ : blue and orange). The first order perturbative prediction (theory) provides a good approximation of learned amplitudes in both 1D and 2D tasks. Shaded area and error bars are 95% CI over 50 seeds.

2005). In this approximation, we assume that the change to the field parameters is small, controlled by the number of fields, and by the large separation between learning rates. Focusing on the place field centers, we derive in App. B the approximation where $\eta_\lambda = 0.0001$ is the learning rate for the field centers and $\eta = 0.01$ is the learning rate for the critic weights:

$$\lambda_i(t) - \lambda_i(0) \approx \frac{\eta_\lambda}{\eta} \left( \frac{2}{\sigma_i^2} + \frac{1}{\sigma_x^2} \right)^{-1} \left[ \frac{\bar{\lambda} - \lambda_i(0)}{\sigma_i^2} + \frac{\bar{\mu}_x - \lambda_i(0)}{\sigma_x^2} \right] w_{v,i}^2(t) \, , \, \eta_\lambda \ll \eta \, , \qquad (7)$$

Under this approximation, each field's center shifts proportionally to the squared magnitude of the critic weights ($w_v^2$), implying that fields at locations with a high value will shift at a faster rate

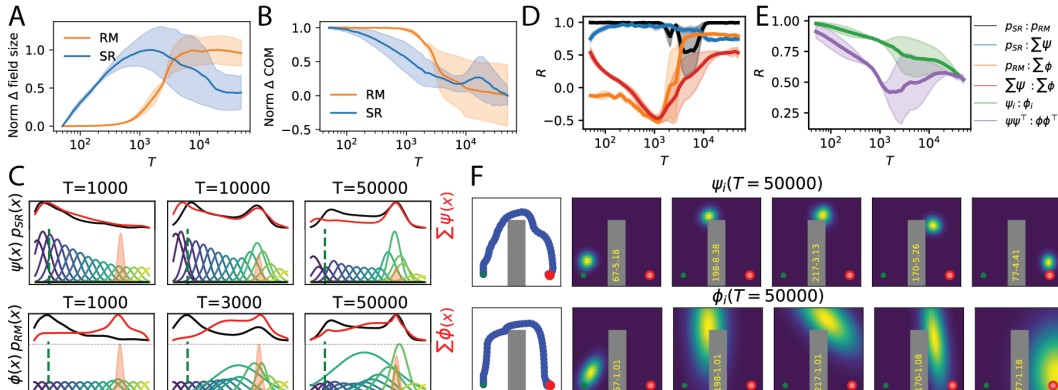

Figure 2: **Reward maximization predicts field enlargement against movement direction, with field dynamics distinct from the successor representation.** (**A-B**) Both Reward Maximization (RM, orange) and Successor Representation (SR, blue) algorithms cause (**A**) field sizes to increase and (**B**) field center of mass to shift backwards against the movement direction when learning in a 1D track, replicating Mehta et al. (1997). Each line shows the average change in an agent initialized with 16 place fields. The change in SR and RM fields were normalized separately to be between 0 to 1 for visualization. (**C**) In the early learning phase ($T = 1000$), both the SR (top row) and RM (bottom row) agents spend a high proportion of time at the start location (black), and learn a policy to spend a higher proportion of time at the target in later phases ($T = 10000, 50000$). The individual SR fields (colored) and SR mean firing rate (red) closely track the proportion of time the agent spends in a location. Conversely, the individual RM fields and mean firing rate show an inverse relationship against the proportion of time the RM agent spends at a location in the early learning phase, but start to align in the later phases. (**D**) The proportion of time SR and RM agents spend at a location is high, positively correlated (black). SR agents show a consistently high, positive correlation (blue) between mean firing rate ($\sum \psi(x)$) and proportion of time spent in a location ($p_{SR}(x)$). Conversely, the correlation between the RM agents' mean firing rate ($\sum \phi(x)$) and time spent at a location ($p_{RM}(x)$) becomes anti-correlated (orange) before becoming positively correlated. Similarly, the SR and RM field densities (red) become anti-correlated before becoming positively correlated at the later learning phase. (**E**) The correlation between the individual field firing rates ($\psi_i(x)$ vs $\phi_i(x)$, green) and the spatial representation similarity matrices ($\boldsymbol{\psi}(x) \cdot \boldsymbol{\psi}(x')$ vs $\boldsymbol{\phi}(x) \cdot \boldsymbol{\phi}(x')$, purple) learned by the SR and RM agents rapidly diverge in the early learning phase but stabilize and become positively correlated in later phases. (**F**) Example change in field size and COM by SR (top row) and RM (bottom row) agents in a 2D arena with an obstacle. Summary statistics in Sup. Fig. 6. The RM agent's field elongation is more pronounced than the SR agent, especially along the trajectory and rotation about the obstacle. Shaded area is 95% CI over 10 seeds.

compared to locations with a low value. In addition to the value of a location, the agent's start location (modeled as a Gaussian with mean $\bar{\mu}_x = -0.75$ and spread $\sigma_x$) and the mean field center location $\bar{\lambda}$ over time under the policy influence each field's displacement. As the reward location is visited frequently, we expect $\bar{\lambda} \approx 0.5$. As the term within the square bracket changes sign depending on the field location, only the fields near the reward location will shift towards the reward, while the rest of the fields will move towards the start location. Due to these influences, the field density at the reward location will increase first followed by a gradual increase in start location (Fig. 1B,E). Additional approximations are needed to model the agent's trajectory and improve the simulation-theory fit for place field centers (App. B). A similar perturbative analysis for amplitudes yields $\alpha_i(t) - \alpha_i(0) \approx 2\frac{\eta_\alpha}{\eta} w_{v,i}^2(t)$ when $\eta_\alpha \ll \eta$, where $\eta_\alpha = 0.0001$ is the learning rate for the $\alpha$ parameters. Thus, fields at locations with a high value will be amplified at a rate similar to the agent learning the value function (Fig. 1F). Therefore, this approximation predicts fields shifting to the start and reward location with field amplification at the reward location.

## 4.2 REWARD MAXIMIZATION RESULTS IN FIELD ENLARGEMENT AGAINST MOVEMENT

We now turn to the next phenomenon where place field sizes increase and their centers shift backward against the movement direction as animals learn to navigate. A proposed account for this phenomenon is that place fields learn to encode future occupancy, that is, given a location $x_t$, the

population of place fields represents the future occupancy probability $p(x_{t+1}|x_t)$ (Stachenfeld et al., 2017). Future occupancy can be learned through Hebbian association of fields that have a correlated firing activity sequence (George et al., 2023; Mehta et al., 2000), or through the successor representation (SR) algorithm, whose objective is to minimize state prediction error by computing a temporal difference error for each place field to learn the transition probabilities (Dayan, 1993; Gardner et al., 2018). Both methods recapitulate field elongation in a 1D track. Here, we show that our reward maximizing (RM) agent does as well.

For comparison purposes, we developed an SR agent that learns the transition probabilities in parallel to policy learning (Sup. Fig. 5A). The SR agent has a similar architecture to our (RM) agent (Fig. 1A), with two key differences: 1) It has one set of place fields with fixed parameters, and only the synapses from these place fields to the actor-critic are optimized for policy learning. 2) There is a separate set of $N$ successor place fields $\psi(x)$ that receive input from the fixed place fields via synapses $U$ which are optimized using the SR algorithm (App. C). We will compare the learned *successor* place fields to the learned place fields in our RM model, following Stachenfeld et al. (2017). We will therefore henceforth refer to the successor place fields simply as place fields.

Both SR and RM agents recapitulate the phenomena seen in (Mehta et al., 1997; Priestley et al., 2022): on average, place fields increase in size over learning (Fig. 2A), and the center of mass (COM) shifts backwards from their initialized positions (Fig. 2B, Sup. Fig. 5C). However, the place fields of the SR and RM agents evolve differently. Both the SR and RM agents initially spend a high proportion of time at the start location and gradually learn a policy to spend a higher proportion of time at the reward location (Fig. 2C). The correlation between the SR and RM agents proportion of time spent in a location is high, positively correlated in most trials (Fig. 2D), except for the decrease between trial 5000 to 10,000 where the RM agent spends a higher proportion of time at the reward location than the SR agent due to faster policy convergence (Sup. Fig. 5B).

The SR, by design, tracks the transition probabilities of the agent's policy. Consequently, the SR mean firing rate $\sum \psi(x)$ closely aligns with the agent's probability of spending time at a location $p_{SR}$, showing a high positive correlation (Fig. 2C, D). Conversely, during early learning, the RM agent exhibits a high mean firing rate $\sum \phi(x)$ at the reward location, which contrasts sharply with the time proportion spent at that location (Fig. 2C), leading to a highly negative correlation between $\sum \phi(x)$ and $p_{RM}$ (Fig. 2D). Interestingly, in the later phase of learning, $\sum \phi(x)$ and $p_{RM}$ become positively correlated. The mean firing rates learned by the SR and RM agents become negatively correlated during the early learning phase but become positively correlated at the later learning phase (Fig. 2D). A similar change in correlation is observed when comparing the individual SR and RM field selectivity or population vectors (Fig. 2E), and the spatial representation similarity matrix (Sup. Fig. 5D) by taking the dot product of SR and RM field firing rates at all locations (Fig. 2E). This demonstrates that both algorithms eventually learn similar spatial representations, but the process of learning these representations are different.

In a 2D arena with an obstacle, both agents show elongation of fields against the agent's direction of movement (Fig. 2, Sup. Fig. 6) while also accounting for the blockage of path by the obstacle. The RM agent shows a significantly larger elongation of fields to span the entire corridor while the elongation of fields by SR is subtle.

### 4.3 STABLE NAVIGATION BEHAVIOR WITH DRIFTING FIELDS

The third phenomena that the model captures has been described as representational drift, where the agent demonstrates stable behavior but the spatial selectivity of individual place fields changes over time (Fig. 3A, Sup. Fig. 8G). Although our agent uses a stochastic policy, both the navigation behavior after 25,000 trials (Fig. 3C, blue) and the population vector (PV) correlation are extremely stable (Fig. 3B, blue). To drive larger variability in the representation, we introduced Gaussian noise to the field parameter updates at every time step (App. D). Increasing the noise magnitude led to a faster decrease in PV correlation but also disrupted agents' policy convergence for magnitudes greater than $10^{-3}$ (Sup. Fig. 7). Hence, we consider the noise magnitudes between $10^{-4}$ and $10^{-3}$. As the noise magnitude increases, agent's reward maximization behavior remains stable while the PV correlation decreases rapidly (Fig. 3B-C). This demonstrates that agents can optimize their policies to maintain stable behavior even though individual spatial selectivity is changing. Interestingly, the spatial representation similarity matrix remains more stable than PV correlation

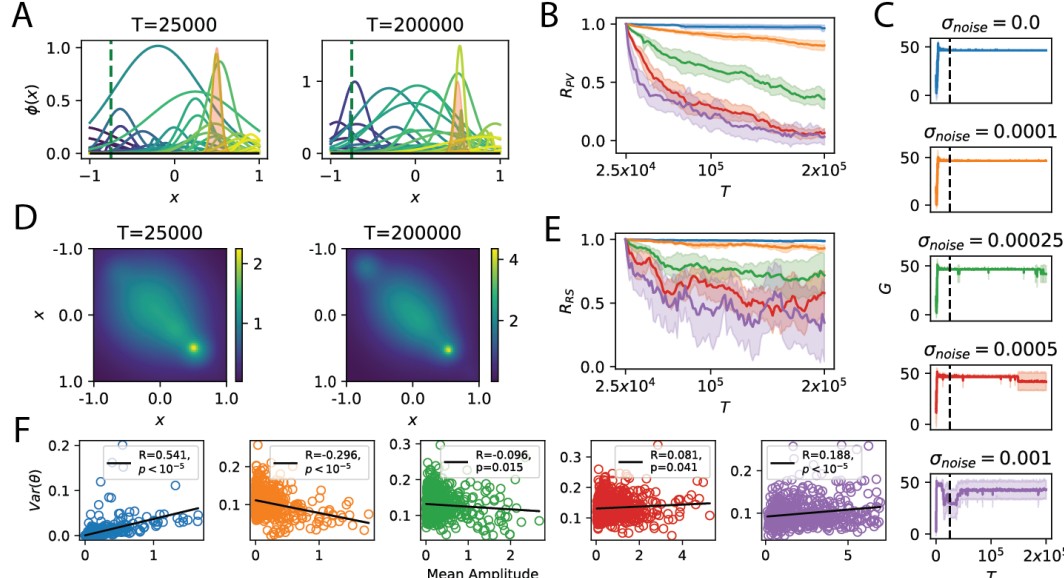

Figure 3: **Stable behavior and representation similarity despite drifting fields.** (A) Injecting Gaussian noise with magnitude $\sigma_{noise} = 0.0001$ into field parameters causes individual field's spatial selectivity to change across trials. (B) Injecting higher noise magnitudes ($\sigma_{noise} = 0.0, 0.0001, 0.0005, 0.001$) leads to a faster decrease in population vector correlation ($R_{PV}$) from trial 25,000 to 200,000. (C) Agents' reward maximization performance ($G$) remains fairly stable when the noise magnitude increases. Beyond $\sigma_{noise} = 0.001$, performance becomes highly unstable. Black dash indicates the trial at which PV and similarity matrix correlation was measured from. (D) The representation similarity matrix (dot product of population activity from (A)) remains stable between trials. (E) With higher noise magnitudes, the similarity matrix correlation ($R_{RS}$) across trials decreases but at a slower rate than PV correlation. (F) Normalized variance in field parameters ($\theta = \{\alpha, \lambda, \sigma\}$) between trials 25,000 to 200,000 quantifies change in individual place fields spatial selectivity. With no noise (blue) or a larger noise magnitude ($\sigma_{noise} = 0.001$), fields with a larger amplitude experiences a greater change in its parameters. When $\sigma_{noise} \in \{0.0001, 0.00025\}$, we see the opposite trend, where fields with a larger amplitude are more stable than fields with a smaller amplitude, replicating Qin et al. (2023). Shaded area is 95% CI over 10 seeds.

(Fig. 3D), even with a higher noise magnitude (Fig. 3E), although the agents are not explicitly optimizing for representational similarity (Qin et al., 2023). Unlike noisy field parameter updates, adding noise to the actor and critic synapses caused the agent's reward maximization behavior, representation similarity correlation and population vector correlation to change at similar rates (Sup. Fig. 7), which is not as consistent with experiment (Sup. Fig. 9 for comparisons to data).

We quantified this drifting behavior at the level of individual neurons by summing the normalized (between $[0, 1]$) variance in each field's parameters ($\sum Var(\tilde{\theta}) = Var(\tilde{\alpha}) + Var(\tilde{\lambda}) + Var(\tilde{\sigma})$) across learning trials, and comparing this against the mean amplitudes for each field. When no Gaussian noise is added (Fig. 3F), fields with a higher mean amplitude showed a higher variance in its parameters, which is expected since fields with a higher amplitude are more likely to be involved in policy learning. Conversely, with a small Gaussian noise, we see the opposite trend where fields with a smaller mean amplitude showed a higher variance in parameters while fields with a higher mean amplitude were more stable. At smaller noise magnitudes, there is a strong positive correlation between higher amplitude fields and the magnitude of actor and critic weights (Sup. Fig. 8). This suggests that high-amplitude fields are more involved in policy learning and thus need stability, whereas less important fields can alter their spatial selectivity, consistent with Qin et al. (2023).

### 4.4 PLACE FIELD REORGANIZATION IMPROVES POLICY CONVERGENCE

As the reward-maximizing model recapitulates experimentally-observed changes in place fields, it is natural to ask what computational advantage these representational changes might offer. To probe the contributions of each field parameter to policy learning, we perform ablation experiments. These

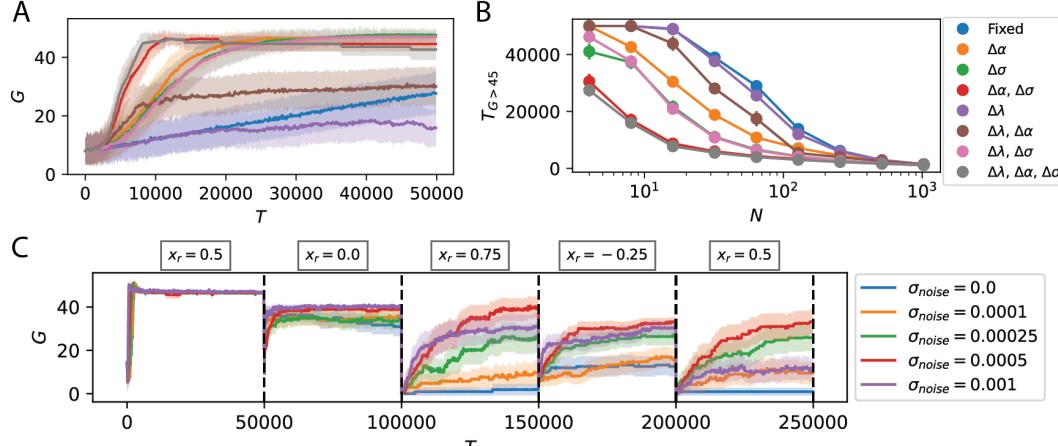

Figure 4: **Field reorganization and noisy updates improve target learning.** **(A)** Optimizing all three field parameters, amplitude, width and center of randomly distributed fields allowed agents ($N = 16, \sigma = 0.1$) to attain the highest cumulative discounted reward ($G$), while fields with fixed field parameters attained the lowest. **(B)** Optimizing place field widths ($\sigma$), followed by field amplitudes ($\alpha$) and lastly field centers ($\lambda$) caused the biggest decrease in the number of trials needed for policy convergence ($T_{G>45}$, attain a running average of $G = 45$ over 300 trials). As the number of fields increased, the number of trials needed for policy convergence decreased and the computational advantage afforded by field optimization extinguished. **(C)** Agents need to navigate to a target that changed after 50,000 trials $x_r = \{0.5, 0.0, 0.75, -0.25, 0.5\}$. Without noisy field parameter updates, agents ($N = 128, \sigma = 0.1$) struggled to learn new targets (blue, $\sigma_{noise} = 0.0$). Field updates with different noise magnitudes influenced the policy convergence speed and maximum cumulative reward for subsequent targets, with $\sigma_{noise} = 0.0005$ (red) demonstrating the highest improvement. Shaded area is 95% CI over 50 seeds.

ablations are particularly important due to the parameter degeneracies in the model: one can trade off the place field amplitudes and the critic and actor weights.

We first considered the task of navigating to a single fixed target. Agents with fixed place fields attained the lowest navigational performance with cumulative reward $G$ plateauing at $G = 33$ per trial (Fig. 4A), and showed the slowest policy convergence even as the number of fields increased (Fig. 4B). Optimizing place field widths ($\sigma$) contributed to the greatest improvement in maximum reward and largest decrease in the number of trials for policy convergence (Fig. 4A-B). Optimizing place field amplitudes ($\alpha$) contributed to the next most significant improvement (Fig. 4A-B). Interestingly, place field center ($\lambda$) optimization did not contribute to a significant improvement in performance, and in fact caused a significant decrease in reward maximization performance and speed of policy convergence when optimized together with the amplitude parameter. Hence, optimizing field widths followed by amplitudes and lastly centers significantly improved agent's reward maximization performance and increased the speed of policy convergence. However, as the number of place fields increase (Fig. 4B), the computational advantage afforded by place field optimization extinguishes. Nevertheless, optimizing all the parameters in a small number of fields, e.g. 8, leads to a similar rate of policy convergence than with a larger number of randomly initialized fields e.g. 128, which hints that representation flexibility could allow efficient learning in systems with few neurons.

We now turn to the influence of noisy fields when learning to navigate to new targets, inspired by Dohare et al. (2024). With the same random field initialization, agents now have to navigate from the same start location to a target that repeatedly changes location. Although all agents learned to navigate to the first and the second targets equally well, agents without noisy field updates struggled to learn the next three targets, and achieved a lower average cumulative reward (Figure 4C). Increasing the noise magnitude led to a monotonic improvement in new target learning. Some fields coding for the initial reward location shifted to code for the new reward location (Sup. Fig. 3). However, noise magnitudes beyond a threshold ($\sigma_{noise} = 0.001$) caused average cumulative reward to decrease. These results suggests that there is a functional role for noise, especially for new target learning. We see a similar improvement in reward maximization performance with noisy field updates in a 2D arena with an obstacle when we either change the target or the obstacle location (Sup. Fig. 12).

## 5    DISCUSSION

We present a two-layer navigation model which uses tunable place fields as feature inputs to an actor and a critic for policy learning. The parameters of the place fields and the policy and value function learn to maximize rewards using the temporal difference (TD) error. Our simple reinforcement learning model reproduces three experimentally-observed neural phenomena: (1) the emergence of a high place field density at rewards, (2) enlargement of fields against the trajectory, and (3) drifting fields without influencing task performance. We analyzed the model to understand how the TD error, number of place fields and noise magnitudes influenced place field representations. Lastly, we demonstrate that learning place field representations with noisy field parameters improves reward maximization and the rate of policy convergence when learning single and multiple targets.

The proposed reinforcement learning model might be a sufficient toy model for theoretical analysis (Bordelon et al., 2024) while remaining biologically grounded enough to make experimentally testable predictions (Kumar et al., 2024a). For instance, our model gives an alternative normative account for field elongation against the trajectory, which can be contrasted with the successor representation algorithm (Kumar et al., 2024b; Raju et al., 2024). As the dynamics of fields are different in these two models, they could be distinguishable by experiments that track fields over the full course of learning (Fig. 2C-E, Sup. Fig. 6). Furthermore, place field width and amplitude optimization increases maximum cumulative reward and accelerates policy convergence (Fig. 4A-B).

Most models that characterized representational drift were not studied under the context of navigational policy learning (Pashakhanloo & Koulakov, 2023; Qin et al., 2023; Ratzon et al., 2024). We showed that increasing the noise magnitudes caused different drift regimes (Fig. 3F; Sup. Fig. 9D), and at very high noise levels navigation behavior started to collapse (Fig. 3C, Sup. Fig. 7). Importantly, we showed that fields in the noisy regime allowed agents to consistently learn new targets in both 1D (Fig. 4C) and 2D (Sup. Fig. 12A-B) environments, without getting stuck in local minima. The biological origins of adding noise to place field parameters can be attributed to noisy synaptic plasticity mechanisms (Attardo et al., 2015; Kappel et al., 2015; Mongillo et al., 2017). Other mechanisms such as unstable dynamics in downstream networks (Sorscher et al., 2023) and modulatory mechanisms such dopamine fluctuations (Krishnan & Sheffield, 2023) could adaptively control drift rates. A difficult experiment that could directly verify our model is to induce or constrain place field drift rates in animals and determine how this perturbation influences new target learning. How fluctuations in dopamine, stochastic actions and stochastic firing rates within place fields drive drift rates needs to be explored. The current model provides a starting point for this investigation.

The proposed model is not without limitations. First, we modeled single peaked place fields instead of the complex representations resulting from single "place" cells, which can be multi-field and multi-scale. Nevertheless, the proposed online reinforcement learning framework is general enough to accommodate other models of place cell description (Mainali et al., 2024; Sorscher et al., 2023)) e.g. Sup. Fig. 14, and can be extended to study representation learning in other brain regions e.g. medial entorhinal (Boccara et al., 2019) or posterior parietal (Suhaimi et al., 2022) cortex. Next, place field parameters are optimized by backpropagating the temporal difference error through the actor and critic components (Sup. Fig. 15). Since the motivation was to develop a normative model whose objective was to maximize rewards, this was a reasonable starting point. However, this model must be extended using biologically-plausible learning rules (Lillicrap et al., 2016; Miconi, 2017; Murray, 2019; Nøkland, 2016) before it can in any way be considered mechanistic (Edelmann & Lessmann, 2018; Kempadoo et al., 2016; Krishnan et al., 2022; Lee et al., 2024; Starkweather & Uchida, 2021). Furthermore, place fields reorganize during latent learning in the absence of rewards. While we have only explored reward maximizing objective, extending our model to examine place field reorganization when optimizing for non-reward based objectives (Fang & Stachenfeld, 2023; Foster et al., 2000; Low et al., 2018) is crucial. Since our model computes gradients using the objective, this should be feasible. While our computational experiments successfully demonstrated the model's effectiveness in reproducing three disparate phenomena, further work should test its robustness across other reinforcement learning algorithms e.g. policy gradient (Kumar & Pehlevan, 2024). Additionally, we need to explore how place field reorganization scales in larger, more complex environments beyond the few 2D environments we considered.

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

# A  DETAILS OF THE PLACE FIELD-BASED NAVIGATION MODEL

## A.1  PLACE FIELDS IN 1D AND 2D ENVIRONMENTS

The agent contains $N$ place fields. In a 1D track, each place field is described as

$$\phi_i(x_t) = \alpha_i^2 \exp\left(-\frac{||x_t - \lambda_i||_2^2}{2\sigma_i^2}\right) , \tag{8}$$

with $\alpha$, $\lambda$ and $\sigma$ describing the amplitude, center and width, adapted from Foster et al. (2000); Kumar et al. (2022; 2024b). Most of the simulations were initialized with amplitudes $\alpha_i = 0.5$ and widths $\sigma_i = 0.1$, with centers uniformly tiling the environment $\boldsymbol{\lambda} = \{-1, ..., 1\}$. Nevertheless, similar representations emerge for amplitudes drawn from a uniform distribution between $[0, 1]$ and widths uniformly drawn between $[0.01, 0.25]$. This parameter initialization was used for ablation studies in Fig. 4. In a 2D arena, each place field is described as

$$\phi_i(x_t) = \alpha_i^2 \exp\left[-\frac{1}{2}(x_t - \lambda_i)^\top \Sigma_i^{-1}(x_t - \lambda_i)\right] , \tag{9}$$

where $\Sigma_i$ is a 2x2 covariance matrix, adapted from Menache et al. (2005). The off-diagonals were initialized as zeros and diagonals initialized to match the variance in the 1D place field description, i.e. $\Sigma_{ii} = 0.1^2$ to ensure field widths are consistent in 1D and 2D.

## A.2  REWARD MAXIMIZATION OBJECTIVE (POLICY GRADIENT)

The objective of the model is to learn a policy $\pi$ parametrized by $W^\pi$ and spatial features $\phi$ parameterized by $\theta$ that maximizes the expected cumulative discounted rewards over trajectories $\tau$ in a finite-horizon setting, modeling the trial structure in neuroscience experiments

$$\mathcal{J}^G = \mathbb{E}_{\tau \sim \phi_\theta, \pi_{W^\pi}}\left[\sum_{t=0}^{T}\sum_{k=0}^{T} \gamma^k r_{t+1+k}\right] = \mathbb{E}\left[\sum_{t=0}^{T} G_t\right] , \tag{10}$$

where $\gamma$ is the discount factor, $r_{t+1}$ is the reward at time step $t + 1$ after choosing an action $g_t$ at time step $t$, and the time horizon $T$ is finite with trials ending after a maximum of 100 steps in the 1D track and 300 steps in the 2D arena.

To maximize the cumulative reward objective, we perform gradient ascent on the policy and place field parameters,

$$\theta_{new} = \theta_{old} + \eta_\theta \nabla_\theta \mathcal{J}^G \quad , \quad W^\pi_{new} = W^\pi_{old} + \eta \nabla_{W^\pi} \mathcal{J}^G , \tag{11}$$

where $\eta_\theta$ and $\eta$ are learning rates for $\theta$ and $W^\pi$ respectively. The gradients are derived using the log-derivative trick,

$$\nabla_{\theta, W^\pi} \mathcal{J}^G = \nabla_{\theta, W^\pi} \mathbb{E}[G(\tau)] \tag{12}$$

$$= \nabla_{\theta, W^\pi} \int_\tau p(\tau|\theta, W^\pi) G(\tau) \tag{13}$$

$$= \int p(\tau|\theta, W^\pi) \nabla_{\theta, W^\pi} \log p(\tau|\theta, W^\pi) G(\tau) \tag{14}$$

$$= \mathbb{E}[\nabla_{\theta, W^\pi} \log p(\tau|\theta, W^\pi) G(\tau)] , \tag{15}$$

where the trajectory $\tau$ describes the state to state transitions. We expand the above using the Markov assumption that the transition to future states depend only on the present state and not on the states preceding it,

$$p(\tau|\theta, W^\pi) = p(x_0) \prod_{t=0}^{T} p(x_{t+1}|x_t) \pi(g_t|x_t; \theta, W^\pi) \tag{16}$$

$$\log p(\tau|\theta, W^\pi) = \log p(x_0) + \sum_{t=0}^{T} (\log p(x_{t+1}|x_t) + \log \pi(g_t|x_t; \theta, W^\pi)) \tag{17}$$

$$\nabla_{\theta, W^\pi} \log p(\tau|\theta, W^\pi) = \sum_{t=0}^{T} \log \pi(g_t|x_t; \theta, W^\pi) . \tag{18}$$

Since the gradients are not dependent on the state transitions, the last line excludes them. Substituting Eq. 18 into Eq. 15 yields

$$\nabla_{\theta,W^\pi} \mathcal{J}^G = \mathbb{E}\left[\sum_{t=0}^{T} \nabla_{\theta,W^\pi} \log \pi(g_t|x_t; \theta, W^\pi) \cdot G_t\right], \tag{19}$$

which completes the full derivation of the policy gradient theorem (Sutton & Barto, 2018; Sutton et al., 1999). The policy gradient objective was used by Kumar & Pehlevan (2024) to optimize the policy and place field parameters. However, this learning signal requires an explicit reward and policy gradient methods are slow to converge as they suffer from high variance due to:

- Monte Carlo sampling: Agents need to sample an entire episode to estimate the expected return $\mathbb{E}_\tau[G_t = r_{t+1} + \gamma r_{t+2} + \gamma^2 r_{t+3} + ...]$ before updating the policy. This can introduce significant variance because the estimate is based on a single path through the stochastic environment, which may not be representative of the expected value over many episodes.

- No Baseline: The basic policy gradient algorithm computes the gradient solely based on the return $G$ from each trajectory. By introducing a baseline (either constant $b$ or dynamically evolving $b_t$ e.g. value function $v_t$), which estimates the expected return from a given state, the variance of the gradient estimate can be reduced, because now the policy learns which action is better than the previous (concept of using an Advantage $A_t$ instead of rewards).

Value based methods (Sutton & Barto (2018), Chapter 3.5) were introduced to address some of these issues. For instance, instead of sampling returns $G_t$, value functions $V_t$ learn to estimate the expected returns

$$V_t = \mathbb{E}[G_t], \tag{20}$$

which can reduce the variance during credit assignment. The combination of policy gradient with value-based methods lead us to the Actor-Critic algorithm.

### A.3 ALTERNATIVE REWARD MAXIMIZATION OBJECTIVE (TEMPORAL DIFFERENCE)

The optimal value function $V_t$ reflects the true expected cumulative discounted rewards, hence the policy gradient objective can be rewritten as

$$\mathcal{J}^G = \mathbb{E}\left[\sum_{t=0}^{T} G_t\right] = \mathbb{E}\left[\sum_{t=0}^{T}\sum_{k=0}^{T} \gamma^k r_{t+1+k}\right] = \sum_{t=0}^{T} V_t, \tag{21}$$

$$= \mathbb{E}\left[\sum_{t=0}^{T} r_{t+1} + \gamma \sum_{k=0}^{T} \gamma^k r_{t+2+k}\right], \tag{22}$$

$$\mathcal{J}^G = \mathbb{E}\left[\sum_{t=0}^{T} r_{t+1} + \gamma G_{t+1}\right] = \mathbb{E}\left[\sum_{t=0}^{T} r_{t+1} + \gamma V_{t+1}\right]. \tag{23}$$

which yields the following self-consistency equation

$$r_{t+1} + \gamma V_{t+1} - V_t \equiv 0, \tag{24}$$

as argued by Frémaux et al. (2013); Sutton & Barto (2018).

Alternatives to policy gradient algorithms propose subtracting a baseline which can be a fixed constant $b$ or a dynamically changing variable $b_t$. Since we have the value function $V_t$ we can modify the objective to be

$$\mathcal{J}^A = \mathbb{E}[G_t - V_t] = \mathbb{E}[A_t] = \mathbb{E}\left[\sum_{t=0}^{T} r_{t+1} + \gamma V_{t+1} - V_t\right], \tag{25}$$

which gives us the Advantage function (Mnih et al., 2016; Schulman et al., 2015). This reduces the variance as the policy has to learn to select actions that gives an advantage over the current value function. We get a learning objective function that is an analogue to maximizing the expected

cumulative discounted returns while subtracting a baseline Eq. 10.

$$\nabla_{\theta,W^\pi} \mathcal{J}^A = \mathbb{E} \left[ \sum_{t=0}^{T} \nabla_\theta \log \pi(g_t|x_t; \theta, W^\pi) \cdot A_t \right] . \tag{26}$$

However, we have assumed that we are given the optimal value function $V_t$ to critique the actor if it is doing better or worse than before. Instead, we can learn to estimate the value function $v_t$ using a critic by minimizing the Temporal Difference error

$$r_{t+1} + \gamma v_{t+1} - v_t = \delta_t . \tag{27}$$

The critic can learn to approximate the true value function by minimizing the mean squared error between the true value function $V_t$ and the predicted $v_t$, or the temporal difference error $\delta_t$

$$\mathcal{L}^v = \mathbb{E} \left[ \sum_{t=0}^{T} \frac{1}{2} \left( V(x_t) - v(x_t; \theta, w^v) \right)^2 \right] \tag{28}$$

$$= \mathbb{E} \left[ \sum_{t=0}^{T} \frac{1}{2} \left( r_{t+1} + \gamma V(x_{t+1}) - v(x_t; \theta, w^v) \right)^2 \right] . \tag{29}$$

Since we do not have the optimal value function $V_t$, we can approximate it by bootstrapping the estimated value function $v_t$ and ensuring that we do not take gradients with respect to the time shifted value estimate $v(x_{t+1})$

$$\mathcal{L}^{TD} = \mathbb{E} \left[ \sum_{t=0}^{T} \frac{1}{2} \left( r_{t+1} + \gamma v(x_{t+1}) - v(x_t; \theta, w^v) \right)^2 \right] \tag{30}$$

$$= \mathbb{E} \left[ \sum_{t=0}^{T} \frac{1}{2} \delta_t^2(\theta, w^v) \right] . \tag{31}$$

We minimize the temporal difference error using gradient descent for the critic to estimate the value function

$$\nabla_{\theta,w^v} \mathcal{L}^{TD} = \frac{\partial \mathcal{L}^{TD}}{\partial \delta} \cdot \frac{\partial \delta}{\partial v} \cdot \nabla_{\theta,w^v} v(\theta, w^v) , \tag{32}$$

$$= \mathbb{E} \left[ \sum_{t=0}^{T} \delta_t \cdot (-1) \cdot \nabla_{\theta,w^v} v(x_t; \theta, w^v) \right] , \tag{33}$$

$$= \mathbb{E} \left[ \sum_{t=0}^{T} -\nabla_{\theta^v} v(x_t; \theta, w^v) \cdot \delta_t \right] . \tag{34}$$

Notice the additional negative sign that pops out when you take the derivative of $\delta$ only with respect to $v_t$

$$\frac{\partial \delta}{\partial v} = \frac{\partial (r_{t+1} + \gamma v_{t+1} - v_t)}{\partial v_t} = -1 , \tag{35}$$

since $r_{t+1}$ and $v_{t+1}$ are treated as constants, we do not take their derivatives. Since we do not have the optimal value function $V_t$ but a biased estimate $v_t$, we can use the temporal difference error as our reward maximization objective

$$\mathcal{J}^{TD} = \mathbb{E} \left[ \sum_{t=0}^{T} r_{t+1} + \gamma v_{t+1} - v_t \right] = \mathbb{E} \left[ \sum_{t=0}^{T} \delta_t \right] . \tag{36}$$

As the value estimation becomes closer to the optimal value $v_t \to V_t$, this objective becomes similar to the advantage objective $\mathcal{J}^{TD} \to \mathcal{J}^A$. Note that we are not directly maximizing the TD error during policy learning. Rather, we want to optimize the policy $\pi$ and place field parameters $\theta$ by gradient ascent, using the biased estimate of the advantage function

$$\nabla_{\theta,W^\pi} \mathcal{J}^{TD} = \mathbb{E} \left[ \sum_{t=0}^{T} \nabla_{\theta,W^\pi} \log \pi(g_t|x_t; \theta, W^\pi) \cdot \delta_t \right] . \tag{37}$$

An alternative interpretation is that during policy learning, the agent learns a policy to maximize the difference between the actual reward and the estimated value

### A.4 COMBINED REWARD MAXIMIZATION OBJECTIVE FOR PLACE FIELD PARAMETERS

In our model (Fig. 1A), actor $W^\pi$ and critic $w^v$ weights are optimized separately, while the place field parameters $\theta$ overlap. The actor uses gradient ascent for Eq. 26, and the critic employs gradient descent for Eq. 34. Since we have a single population of place fields, we optimize these parameters to support both objectives. Thus, we derive a combined objective function to update $W^\pi$, $w^v$, and $\theta$ in a single gradient pass

$$\nabla_{W^\pi, w^v, \theta} \mathcal{J} = \nabla_{W^\pi, w^v, \theta} \mathcal{J}^{TD} - \nabla_{W^\pi, w^v, \theta} \mathcal{L}^{TD} \tag{38}$$

$$= \mathbb{E}\left[\sum_{t=0}^{T} \nabla_{W^\pi, w^v, \theta} \log \pi(g_t|x_t; W^\pi, \theta)\delta_t\right] - \mathbb{E}\left[\sum_{t=0}^{T} -\nabla_{W^\pi, w^v, \theta} v(x_t; w^v, \theta)\delta_t\right], \tag{39}$$

$$= \mathbb{E}\left[\sum_{t=0}^{T} \nabla_{W^\pi, w^v, \theta} \log \pi(g_t|x_t; W^\pi, \theta)\delta_t + \nabla_{W^\pi, w^v, \theta} v(x_t; w^v, \theta)\delta_t\right], \tag{40}$$

$$= \mathbb{E}\left[\sum_{t=0}^{T} \left(\nabla_{W^\pi, w^v, \theta} \log \pi(g_t|x_t; W^\pi, \theta) + \nabla_{W^\pi, w^v, \theta} v(x_t; w^v, \theta)\right)\delta_t\right]. \tag{41}$$

where $\nabla_{w^v} \mathcal{J}^{TD} = 0$ and $\nabla_{W^\pi} \mathcal{L}^{TD} = 0$ since the respective objectives are not parameterized by $w^v$ and $W^\pi$ respectively. This means that $W^\pi$ is tuned to maximize $\mathcal{J}^{TD}$, $w^v$ is tuned to minimize $\mathcal{L}^{TD}$ and $\theta$ is tuned to balance both the objectives.

Since most optimizers e.g. in Tensorflow, PyTorch perform gradient descent, not ascent, we can minimize the negative policy gradient Eq. 26, which is equivalent to the negative log likelihood

$$\nabla_{W^\pi, w^v, \theta} \mathcal{L} = -\nabla_{W^\pi, w^v, \theta} \mathcal{J}^{TD} + \nabla_{W^\pi, w^v, \theta} \mathcal{L}^{TD} \tag{42}$$

$$= -\mathbb{E}\left[\sum_{t=0}^{T} \nabla_{W^\pi, w^v, \theta} \log \pi(a_t|x_t; W^\pi, \theta) \cdot \delta_t\right] + \mathbb{E}\left[\sum_{t=0}^{T} -\nabla_{W^\pi, w^v, \theta} \tilde{v}(x_t; w^v, \theta) \cdot \delta_t\right], \tag{43}$$

$$= \mathbb{E}\left[\sum_{t=0}^{T} \nabla_{W^\pi, w^v, \theta} -\log \pi(a_t|x_t; W^\pi, \theta) \cdot \delta_t\right] + \mathbb{E}\left[\sum_{t=0}^{T} -\nabla_{W^\pi, w^v, \theta} \tilde{v}(x_t; w^v, \theta) \cdot \delta_t\right], \tag{44}$$

$$= \nabla_{W^\pi, w^v, \theta} \mathcal{L}_\pi^{TD} + \nabla_{W^\pi, w^v, \theta} \mathcal{L}_v^{TD}. \tag{45}$$

which is the same update rule used in Mnih et al. (2016); Wang et al. (2018) to train the actor and critic separately while the feature parameters are trained jointly.

It is also possible to initialize two separate populations of place fields, each for the actor and critic. Alternatively, we only optimize place field parameters using the actor's objective while the critic uses the spatial features to learn the value function. The converse is also possible where the place field parameters and critic weights are optimized to minimize the TD error while the actor learns a policy without optimizing the spatial representations, as we did in the perturbative approximation (App. B). From numerical experiments, optimizing place field parameters using both the actor and critic objectives allowed the agent to achieve the fastest policy convergence and highest cumulative reward performance (Sup. Fig. 15).

### A.5 ONLINE UPDATE OF PLACE FIELD AND ACTOR-CRITIC PARAMETERS

Now, we derive an online implementation of Eq. 6 which is the same as Eq. 41, so that all parameters are updated at every time step. Extending from Foster et al. (2000); Kumar et al. (2022), the actor

and critic weights are updated according to the gradients

$$\Delta w_i^v(t+1) = \eta \delta_t \phi_i(x_t) \quad , \quad \Delta W_{ji}^\pi(t+1) = \eta \delta_t \phi_i(x_t) \tilde{g}_{t,j}^\top , \tag{46}$$

where $\tilde{g}_{t,j} = g_t - P$ and $\eta = 0.01$. The gradient updates for place field parameters follow

$$\Delta \theta_i(t+1) = \eta_\theta \delta_t \left( w_i^v(t) + W_{ji}^\pi(t) \cdot \tilde{g}_{t,j} \right) \nabla_\theta \phi_i(x_t; \theta_i) , \tag{47}$$

where we use a significantly smaller learning rate $\eta_\theta = 0.0001$ so that the spatial representation evolves in a stable manner. Specifically, each field parameter is updated according to

$$\delta_{i,t}^{bp} = \delta_t \left( w_i^v(t) + W_{ji}^\pi(t) \cdot \tilde{g}_{t,j} \right) , \tag{48}$$

$$\Delta \alpha_{i,t} = \eta_\alpha \cdot \delta_{i,t}^{bp} \cdot \phi_i(x_t) \cdot \left( \frac{2}{\alpha_i} \right) , \tag{49}$$

$$\Delta \lambda_{i,t} = \eta_\lambda \cdot \delta_{i,t}^{bp} \cdot \phi_i(x_t) \cdot \left( \frac{x_t - \lambda_i}{\sigma_i^2} \right) , \tag{50}$$

where $\delta_{i,t}^{bp}$ is the TD error gradient that has been backpropagated through the actor and critic weights. Using just the $w_i^v(t)$ or $W_{ji}^\pi$ weights alone to backpropagate the TD error influences the representation learned by the place field population and ultimately the navigation performance (Sup. Fig. 15).

There are two ways to optimize the place field width parameter. The first and straightforward method is to update the width parameter according to

$$\Delta \sigma_{i,k,t} = \eta_\sigma \cdot \delta_{i,t}^{bp} \cdot \phi_{i,k}(x_t) \cdot \left( \frac{(x_t - \lambda_i)^2}{\sigma_{i,k}^3} \right) , \tag{51}$$

where $k = 1$ in a 1D place field. In a 2D place field with $k = 2$, we can update the diagonal elements in the 2D matrix while keeping the off-diagonals to zeros as in Menache et al. (2005). However, fields will only elongate along each axis. Instead, in our simulations, we optimized the off-diagonals using the same gradient flow equations. However, we needed to include additional constraints so that each place field's covariance matrix remains 1) symmetric, 2) bounded, and 3) positive semi-definite to perform matrix inversion. Specifically, the covariance matrix was bounded between $[10^{-5}, 0.5]$ to prevent exploding widths and gradients.

## B DERIVATION FOR PERTURBATIVE EXPANSION

The dynamics of place field parameters are nonlinear and difficult to characterize analytically. To gain some analytical tractability, we impose a strong separation of timescales between policy learning updates and place field parameter updates. To do so, we set the learning rates for the actor-critic $\eta$ to be much larger than the learning rates for the place field parameters $\eta_\alpha, \eta_\lambda, \eta_\sigma \ll \eta$. In simulations, we use $\eta = 0.01$ and $\eta_\theta = 0.0001$.

The critic estimates the value as

$$v(x_t) = \sum_{i=1}^{N} w_i \phi_i(x_t, \boldsymbol{\theta}_i), \tag{52}$$

where $\boldsymbol{\theta}_i = (\alpha_i, \lambda_i, \sigma_i)$ are neuron specific parameters (amplitude, mean, and bandwidth respectively). We write $w^v$ as $w$ for clarity. To start with let's just consider

$$\phi_i(x_t, \boldsymbol{\theta}_i) = \alpha_i^2 \exp\left(-\frac{1}{2\sigma_i^2}(x_t - \lambda_i)^2\right). \tag{53}$$

We consider a TD based update, which in the gradient flow (infinitesimal learning rate) limit can be approximated as

$$\frac{d}{dt}\boldsymbol{w}(t) = \boldsymbol{M}(t)(\boldsymbol{w}^V - \boldsymbol{w}(t)), \tag{54}$$

$$\frac{d}{dt}\boldsymbol{\theta}_i(t) = \epsilon\, w_i(t)\mathbb{E}_{x_t}\nabla_{\boldsymbol{\theta}_i}\phi_i(x_t, \boldsymbol{\theta}_i)\delta_t, \tag{55}$$

The key assumption we make is that the dimensionless ratio of learning rates, $\epsilon$ is perturbatively small

$$\epsilon = \frac{\eta_\theta}{\eta} \ll 1, \tag{56}$$

where $\eta_\theta$ is the learning rate for the place field parameters $\boldsymbol{\theta}_i$ and $\eta$ is the learning rate for the actor-critic. The matrix $\boldsymbol{M}(t) = \boldsymbol{\Sigma}(t) - \gamma\boldsymbol{\Sigma}_+(t)$ where $\boldsymbol{\Sigma} = \langle\boldsymbol{\psi}(x_t)\boldsymbol{\psi}(x_t)\rangle$ and $\boldsymbol{\Sigma}_+(t) = \langle\boldsymbol{\psi}(x_t)\boldsymbol{\psi}(x_{t+1})^\top\rangle$ depends on the equal time and time-step shifted correlations of features. The vector $\boldsymbol{w}^V = \boldsymbol{M}^{-1}\boldsymbol{\Sigma}\boldsymbol{w}_R$ where $\boldsymbol{w}_R \cdot \boldsymbol{\psi}(x) = R(x)$. We investigate a simple perturbation series.

$$\boldsymbol{w}(t) = \boldsymbol{w}_0(t) + \epsilon\boldsymbol{w}_1(t) + \epsilon^2\boldsymbol{w}_2(t) + \dots$$

$$\boldsymbol{\theta}(t) = \boldsymbol{\theta}_0(t) + \epsilon\boldsymbol{\theta}_1(t) + \epsilon^2\boldsymbol{\theta}_2(t) + \dots \tag{57}$$

and examine the dynamics up to first order in $\epsilon$. We will show that this recovers many qualitative features of the observed representational updates.

The leading zeroth order dynamics are

$$\frac{d}{dt}\boldsymbol{\theta}_0(t) = 0, \quad \frac{d}{dt}\boldsymbol{w}_0(t) = \boldsymbol{M}_0(\boldsymbol{w}_V - \boldsymbol{w}_0(t)), \tag{58}$$

where $\boldsymbol{M}_0 = \boldsymbol{\Sigma}(0) - \gamma\boldsymbol{\Sigma}_+(0)$ is the initial feature covariance under the initial policy.

### B.1 PLACE FIELD AMPLITUDE

We start by asserting a separation of timescales between training readout weights and feature parameters during a simple TD learning setup

$$\frac{d}{dt}w_i(t) = \sum_j M_{ij}(w_j^V - w_j), \tag{59}$$

$$\frac{d}{dt}\alpha_i(t) = \epsilon\,\frac{2}{\alpha_i(t)}w_i\sum_j M_{ij}(w_j^V - w_j), \tag{60}$$

The zero-th order solution to Eq. 54 is

$$\Delta\boldsymbol{w}_0(t) \equiv \boldsymbol{w}_V - \boldsymbol{w}_0(t) = \exp(-\boldsymbol{M}t)\,\boldsymbol{w}_V, \tag{61}$$

$$\boldsymbol{w}_0(t) = [\boldsymbol{I} - \exp(-\boldsymbol{M}t)]\boldsymbol{w}_V, \tag{62}$$

which can be substituted in to get the first order correction to the dynamics for $\theta$

$$\frac{d}{dt}\boldsymbol{\alpha}_1(t) = 2\boldsymbol{\alpha}_0^{-1} \odot [\boldsymbol{I} - \exp(-\boldsymbol{M}t)]\boldsymbol{w}_V \odot \boldsymbol{M}\exp(-\boldsymbol{M}t)\,\boldsymbol{w}_V\,. \tag{63}$$

Under the condition that $\boldsymbol{\alpha}_0 = \mathbf{1}$ and $\boldsymbol{M} = \boldsymbol{M}^\top$ we can work out an exact expression in terms of the eigendecomposition $\boldsymbol{M} = \sum_k \lambda_k \boldsymbol{u}_k \boldsymbol{u}_k^\top$

$$\boldsymbol{\alpha}_1(t) = 2\sum_{k\ell}(\boldsymbol{w}_V \cdot \boldsymbol{u}_k)(\boldsymbol{u}_\ell \cdot \boldsymbol{w}_V)\,(\boldsymbol{u}_k \odot \boldsymbol{u}_\ell)\left[(1 - e^{-\lambda_k t}) - \frac{\lambda_k}{\lambda_k + \lambda_\ell}(1 - e^{-(\lambda_k + \lambda_\ell)t})\right]\,, \tag{64}$$

we can approximate this at late times as

$$\lim_{t \to \infty}\boldsymbol{\alpha}_1(t) \approx 2\boldsymbol{w}_V \odot \boldsymbol{w}_V\,. \tag{65}$$

As $t \to \infty$ we can approximate this as $\lim_{t \to \infty}\boldsymbol{\theta}(t) \approx 2(\boldsymbol{w}_V)^2$. This indicates that neurons which are heavily involved in the reproduction of the value function are upweighted in their amplitude.

## B.2 FIELD CENTER

Based on the place field center update equation and rewriting the terms as above,

$$\frac{d}{dt}\lambda_i(t) \approx \epsilon\,\frac{x_t - \lambda_i}{\sigma_i^2}\,w_i\phi_i(x)\sum_j \phi_j(x)(w_j^V - w_j)\,. \tag{66}$$

We need to compute an average over spatial positions. We approximate the space position early in training as a Gaussian with mean $s_0$ and variance $\sigma_x^2$

$$\left\langle \frac{(x_t - \lambda_i)}{\sigma^2}\phi_i(x)\phi_j(x)\right\rangle \approx \frac{\mu_{ij} - \lambda_i}{\sigma^2}M_{ij}\,, \tag{67}$$

where $\mu_{ij} = \left(\frac{2}{\sigma^2} + \frac{1}{\sigma_x^2}\right)^{-1}\left(\frac{1}{\sigma^2}(\lambda_i + \lambda_j) + \frac{1}{\sigma_x^2}\bar{\mu}_x\right)$ is the mean value of $x$ obtained by the above Gaussian integral under the approximation that $p(x) \sim \mathcal{N}(\bar{\mu}_x, \sigma_x^2)$. Approximating $\lambda_j$ as the mean position of the tuning curves $\bar{\lambda}$ we obtain the following prediction

$$\boldsymbol{\lambda}(t) - \boldsymbol{\lambda}(0) \approx \epsilon\boldsymbol{w}^V \odot \left[\left(\frac{2}{\boldsymbol{\sigma^2}} + \frac{1}{\sigma_x^2}\right)^{-1}\left(\frac{1}{\boldsymbol{\sigma^2}}(\boldsymbol{\lambda}(0) + \bar{\lambda}) + \frac{1}{\sigma_x^2}\bar{\mu}_x\right) - \boldsymbol{\lambda}(0)\right] \odot [\boldsymbol{I} - \exp(-\boldsymbol{M}t)]\,\boldsymbol{w}^V\,. \tag{68}$$

Following the solution in Eq. 62, we can approximate this at late times as

$$\lim_{t \to \infty}\boldsymbol{\lambda}(t) - \boldsymbol{\lambda}(0) \approx \epsilon\boldsymbol{w}^V \odot \left[\left(\frac{2}{\boldsymbol{\sigma^2}} + \frac{1}{\sigma_x^2}\right)^{-1}\left(\frac{1}{\boldsymbol{\sigma^2}}(\boldsymbol{\lambda}(0) + \bar{\lambda}) + \frac{1}{\sigma_x^2}\bar{\mu}_x\right) - \boldsymbol{\lambda}(0)\right] \odot \boldsymbol{w}^V\,. \tag{69}$$

Hence, in addition to the value of a location, three additional factors influence each field's displacement.

$$\lambda_i(t) - \lambda_i(0) \approx \frac{\eta_\lambda}{\eta}\left(\frac{2}{\sigma_i^2} + \frac{1}{\sigma_x^2}\right)^{-1}\left[\frac{\bar{\lambda} - \lambda_i(0)}{\sigma_i^2} + \frac{\bar{\mu}_x - \lambda_i(0)}{\sigma_x^2}\right]w_{v,i}^2(t)\,, \quad \eta_\lambda \ll \eta\,, \tag{70}$$

where $\bar{\lambda}$ is the agent's expected location sampled from its policy, $\bar{\mu}_x = -0.75$ is the starting location and $\sigma_x$ is the estimated spread of the trajectory. This analysis suggests that fields will be influenced by both the start location and the location where the agent spends a higher proportion of time at. In later learning phases, this will be the reward location $\bar{\lambda} = 0.5$. Consequently, only the fields near the reward location will shift towards the reward, while the rest of the fields will move towards the start location. We illustrate this perturbative approximation at early and late times of training in Figure 5. The theory is quite accurate early in training, but fails at sufficiently long training time.

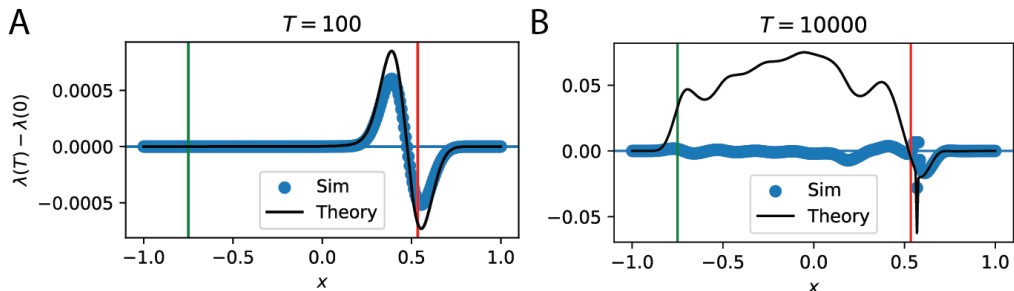

Figure 5: **Difference in early versus late time perturbative approximation.** Blue scatter points shows the magnitude and direction of change in ($N = 256$) field center position compared to the position at which the fields were initialized ($\lambda_i(T) - \lambda_i(0)$). **(A)** In early time, the perturbative expansion is a good fit to the field center displacement, and captures the shift in fields towards the reward location $x_r = 0.5$ (red) **(B)** As learning proceeds, the approximation begins to break down for fields further from the reward location. Free parameters were fit with $\bar{\lambda} = 0.535$ and $\sigma_x = 0.45$.

## C  DETAILS FOR THE SUCCESSOR REPRESENTATION AGENT

The generalized temporal difference error is given by

$$\delta_{t,j}^{SR} = \phi_j(x_t) + \gamma \psi_j^\pi(x_{t+1}) - \psi_j^\pi(x_t)\,, \tag{71}$$

with $M_i$ representing the predicted successor representation and $\phi(x)$ representing the initialized place field representation that is not optimized.

$$\psi_i^\pi(x_t) = \sum_i^N [U_{ji}]_+ \phi_i(x_t)\,, \tag{72}$$

The successor representation is computed using a summation of the place fields with a learned matrix $U$ that is positively rectified. The rectification is necessary to have a non-negative representation.

$$\Delta U_t = \phi_i(x_t) \cdot \delta_{t,j}^\top\,, \tag{73}$$

The matrix $U$ is initialized as an identity matrix and is updated using a two-factor rule using the TD error as in Gardner et al. (2018).

## D  DETAILS FOR NOISY FIELD UPDATES

To induce drift, we independently introduced noise to field amplitudes, centers and width, as well as the synapses to the actor and critic ($\theta \in \{\alpha, \lambda, \sigma, w^v, W^\pi\}$).

$$\theta_{t+1} = \theta_t + \xi_t\,, \tag{74}$$

where the noise term $\xi_t$ are independent Gaussian noises with zero mean and magnitude $\sigma_{noise} \in \{10^{-6}, 10^{-1}\}$. We performed a noise sweep to determine how increasing the noise magnitude affected the agent's reward maximization behavior, population vector correlation and representation similarity. Refer to Sup. Fig. 7.

# E  SUPPLEMENTARY FIGURES

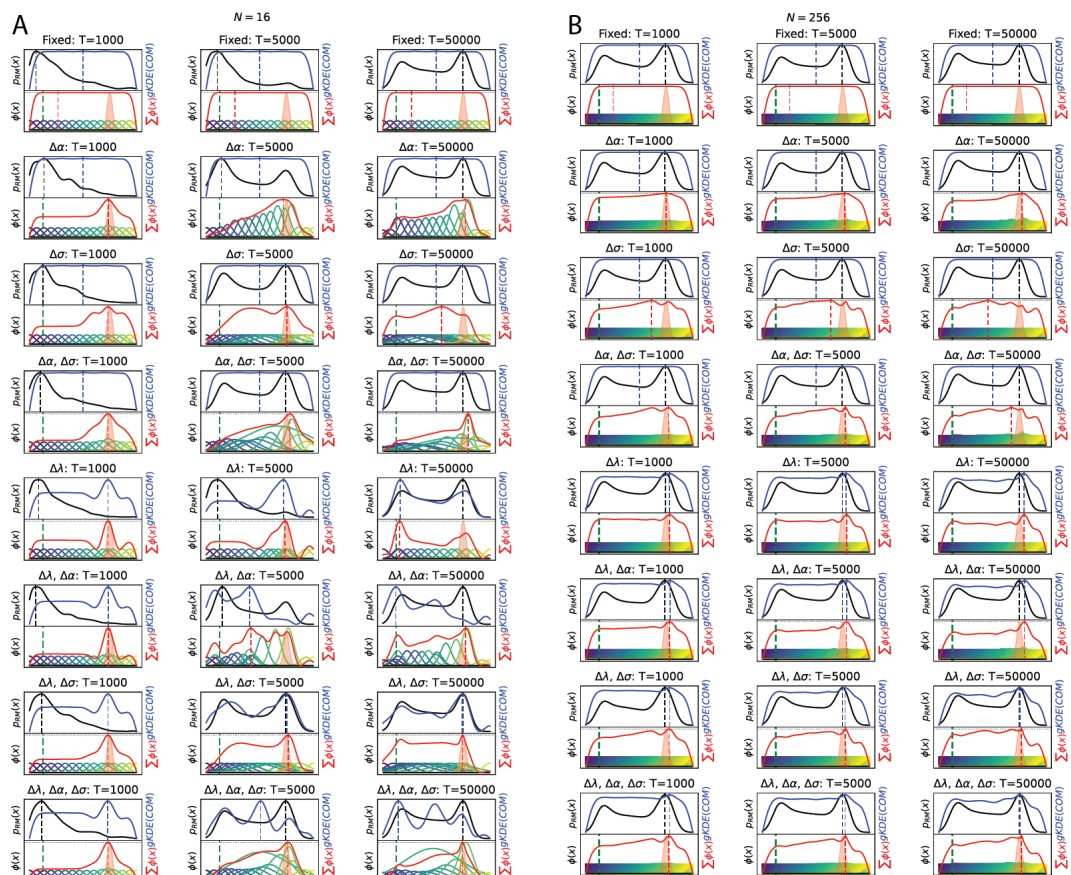

Supplementary Figure 1: **Influence of place field parameter optimization for a single seed.** Example change in individual field's spatial selectivity ($\phi(x)$, colored), mean firing activity at a location ($\sum_i^N \phi_i(x)$), field density which is the number of Center of Mass (COM) in a location after smoothing with a Gaussian kernel density estimate (gKDE) ($gKDE(COM)$, blue) and, the frequency of being in a location ($p_{RM}(x)$), when optimizing different combinations of field parameters ($\alpha, \lambda, \sigma$) during reward maximization (RM). The location in which the highest value for mean firing activity, field density and frequency is attained is indicated by a red, blue and black vertical dash line respectively. Optimizing a **(A)** small number ($N = 16$) and **(B)** large number of place fields yields a similar high mean firing rate at the reward location followed by the start location. However, the field density evolves differently when in the low field regime, **(A)** a high density emerges at the reward location in the early stages of learning, but it shifts to the start location at later stages of learning. **(B)** In the high field regime, a high field density at the reward location remains stable throughout learning. Note that COM changes only when the place field centers are optimized ($\Delta\lambda$). Distribution is shown for a single seed run for a homogeneous place field population that has been initialized by with equal spacing between field centers ($\lambda \in [-1, 1]$), equal amplitude ($\alpha = 0.5$) and width ($\sigma = 0.01$).

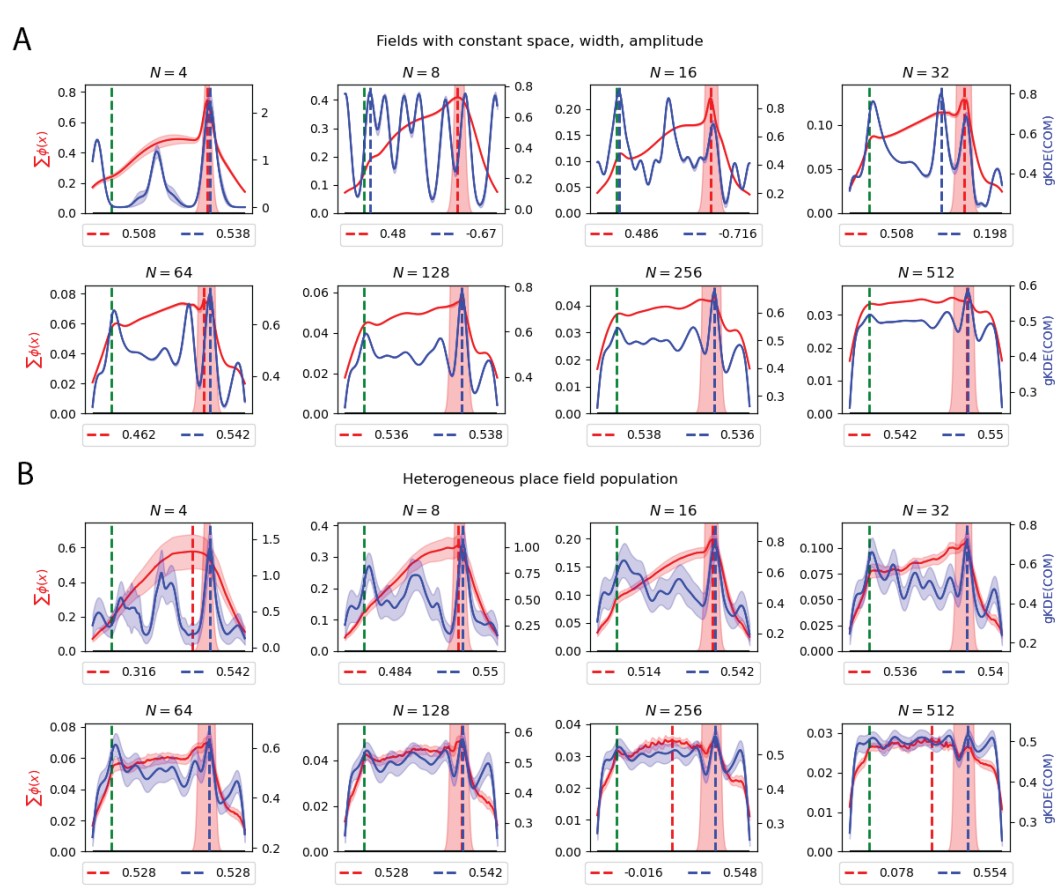

Supplementary Figure 2: **Average change in field density and mean firing rate for different number of place fields.** Vertical blue and red dash lines indicate the location with the highest density and mean firing rate, with the legend indicating the location ($x$). **(A)** Homogeneous place field distribution was initialized with field parameters similar to Sup. Fig. 1, equal spacing between field centers ($\lambda \in [-1, 1]$), equal amplitude ($\alpha = 0.5$) and equal width ($\sigma = 0.01$). **(B)** All place field parameters center ($\lambda$), amplitude ($\alpha$), and width ($\sigma$) were initialized by sampling from a uniform distribution between $[-1, 1]$, $[0, 1]$, $[10^{-5}, 0.1]$ respectively to model heterogeneous place field population. Learning rates for the place field parameters and actor-critic were $n_\theta = 0.0001$ and $n = 0.01$ respectively. Shaded area is 95% CI over 50 different seeds.

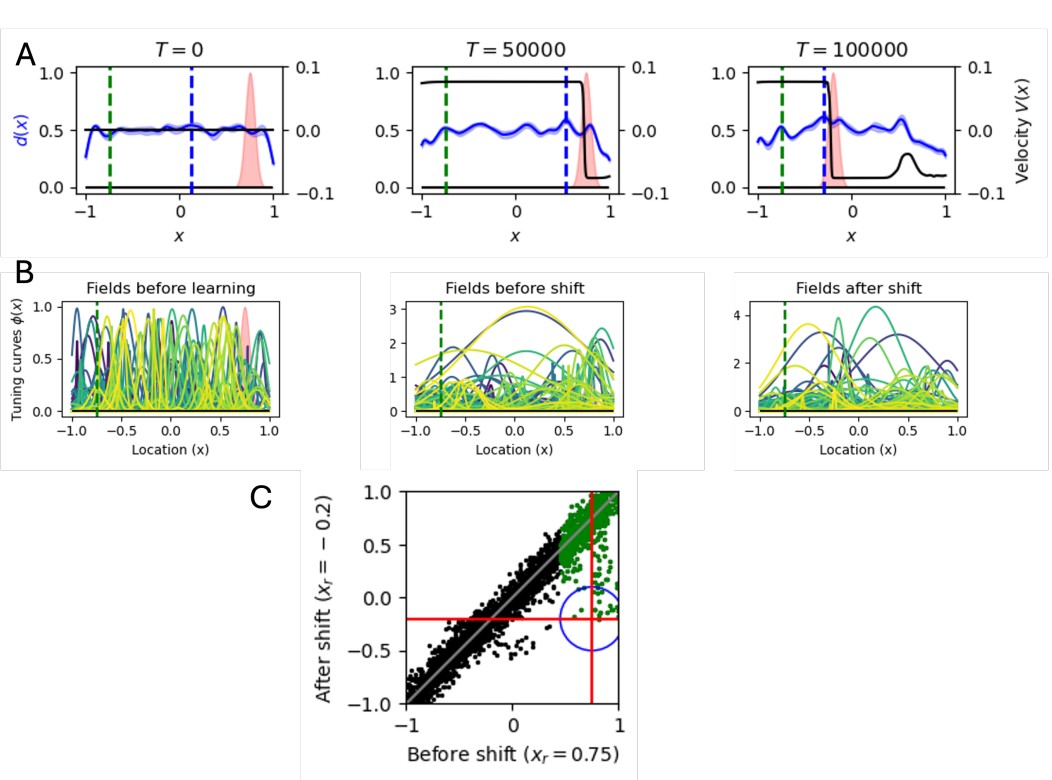

Supplementary Figure 3: **A small proportion of reward-encoding place fields shift to the new reward location.** Agents with $N = 256$ place fields and Gaussian noise injected to field parameters ($\sigma_{noise} = 0.0001$) were trained to navigate to a reward location at $x_r = 0.75$ for 50,000 trials, thereafter the reward location was shifted to $x_r = -0.2$ for the next 50,000 trials. **(A)** Place field density at the start of learning was uniformly distributed (left) and increased near the first reward location at the end of the first 50,000 trials (center). After the shift in reward location, a high density of fields emerged at the new reward location (right). The black line shows the learned policy, where a velocity of 0.1(-0.1) indicates moving right (left). Agents learn to navigate to the reward location, both before and after the shift. **(B)** Example distribution of individual place fields before learning (left), before the shift (center) and after the shift (right). All place field parameters $\lambda$, $\alpha$, and $\sigma$ were initialized by sampling from a uniform distribution between $[-1, 1]$, $[0, 1]$, $[10^{-5}, 0.1]$ respectively to model heterogeneous place field population. Notice the backward shift of some place fields that were at the initial reward location to the new reward location. **(C)** About 2.6% of the place fields coding for the initial reward at $x_r = 0.75$ (green dots) shifted to the new reward location at $x_r = -0.2$ (about 19 of the 734 green dots are within the blue circle). Other place fields at $x_r = -0.2$ increased their firing rate to encode the new reward location. We see a large number of fields shifting backward, though not entirely to the new reward location. Shaded area shows 95% CI for 10 seeds of agents with 256 place fields each. Black and green dots show a total 2560 place fields for all 10 agents.

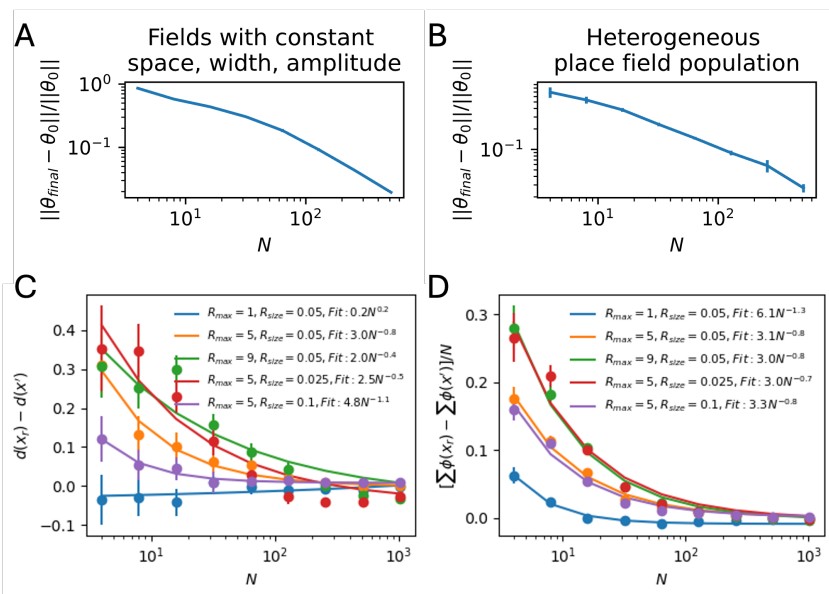

Supplementary Figure 4: **Weak feature learning with large number of place fields.** Critic $w_i^v$ and actor $W_{ji}^\pi$ weights were initialized by sampling from a random normal distribution $\mathcal{N}(0, 10^{-5})$, despite the number of place fields $N$, similar to Foster et al. (2000); Frémaux et al. (2013); Kumar et al. (2022); Zannone et al. (2018). **(A)** Homogeneous place field population: Place field parameters were initialized with equal spacing between field centers ($\lambda \in [-1, 1]$), equal amplitude ($\alpha = 0.5$) and equal width ($\sigma = 0.01$). **(B)** Heterogeneous place field population: All place field parameters center ($\lambda$), amplitude ($\alpha$), and width ($\sigma$) were initialized by sampling from a uniform distribution between $[-1, 1]$, $[0, 1]$, $[10^{-5}, 0.1]$ respectively. **(A-B)** The sum of the L2 norm for each place field's center $\lambda$, amplitude $\alpha$ and width $\sigma$ between its initialized and final value decreases as the number of fields available increases. Hence, as the number of fields increases, the change in each place field's parameter becomes smaller. This suggests a weak feature learning regime with large N. **(C)** Similar to Fig. 1D. Density at the reward location $d(x_r)$ compared to non-reward location $d(x')$ decreases with a higher number of fields. **(D)** The mean firing rate at the reward location $\sum \phi(x_r)$ compared to non-reward location $\sum \phi(x')$ decreases with a higher number of fields. **(C-D)** Density and mean firing rate at the reward location are proportional to the reward magnitude ($R_{max}$), and inversely proportional to the size of the reward location ($R_{size}$). Error bars show 95% CI over 50 different seeds.

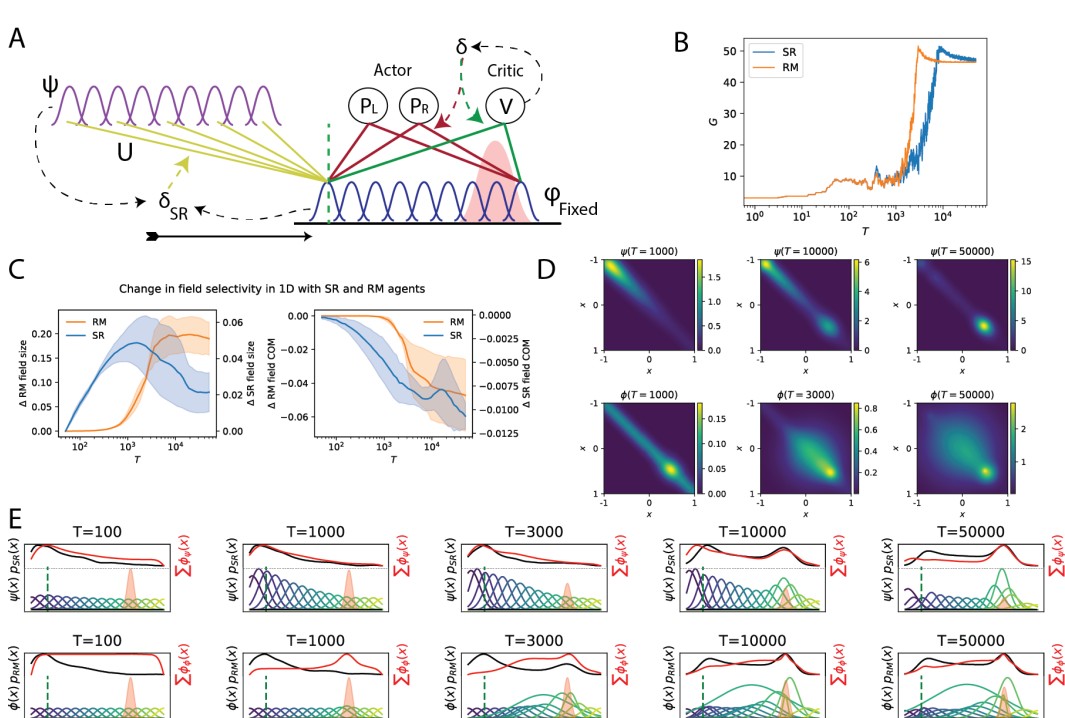

Supplementary Figure 5: **SR agent architecture and field dynamics. (A)** Successor Representation (SR) agent architecture to learn a navigational policy and the SR place fields. Only the synapses from the initialized place field ($\phi_{fixed}$) to the actor (red) and critic (green), and the synapses ($U$) to the SR fields ($\psi$) were plastic. Refer to App. C for implementation details. **(B)** Difference in reward maximization behavior between SR and RM agent, contributing to the dip in correlation between the proportion of time spent in a location by both agents in Fig. 2D black line. **(C)** Average change for 16 place fields' size (firing rate greater than $10^{-3}$ in the track) (left) and center of mass (right) when SR and RM agents navigate in a 1D track with the absolute change reflected in the left and right y axis. Shaded area shows 95% CI over 10 seeds. **(D)** Spatial representation similarity matrix for SR (top row) and RM (bottom row) agents in a 1D track is visualized by taking the dot product of the place field activity at each location. (E) Change in individual place field's spatial selectivity (colored), mean firing rate (red) and frequency of being in a location (black) when fields are learned using the Successor Representation (Top row) and Reward Maximizing objective (Bottom row). Top panels T=1000, 10,000 and 50,000 were selected for SR and bottom panels T=1000, 3000 and 50,000 were selected for RM in the paper due to space constraints.

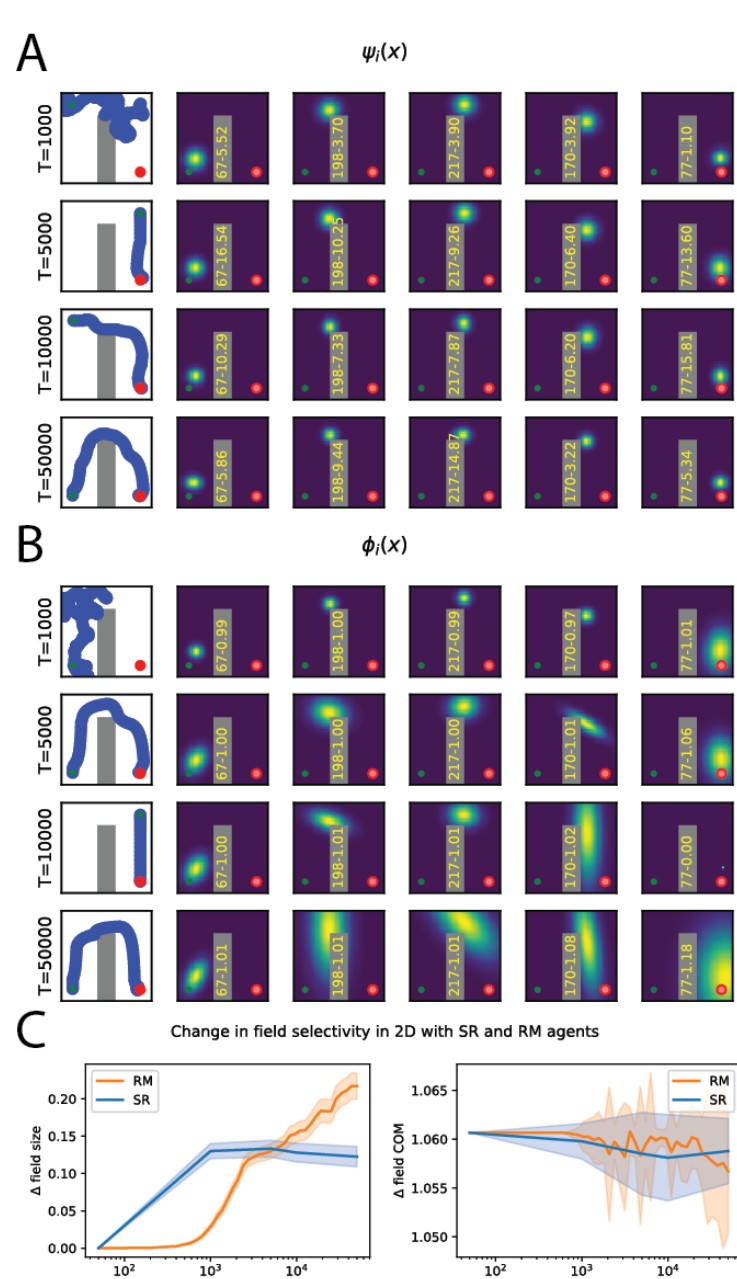

Supplementary Figure 6: **Field elongation in 2D arena.** **(A-B)** 2D Place field distortion dynamics by SR **(A)** and RM **(B)** agents as learning proceeds. Numbers in yellow on the obstacle indicates (Field ID)-(Maximum firing rate). **(C)** Average change in 256 field sizes (left) and center of masses (right) for SR and RM agents navigating in a 2D arena. Shaded area shows 95% CI over the 256 fields. Note that agent start randomly from three different locations $x_{start} \in \{(-0.75, -0.75), (-0.75, 0.75), (0.75, 0.75)\}$ to navigate to the target at $x_r = (0.75, -0.75)$. The change in field COM shows the average change in center of mass with respect to each starting location. Hence, the averaged backward shift in center of mass might not be extensive. Refer to Fig. 1C for change in goal representation.

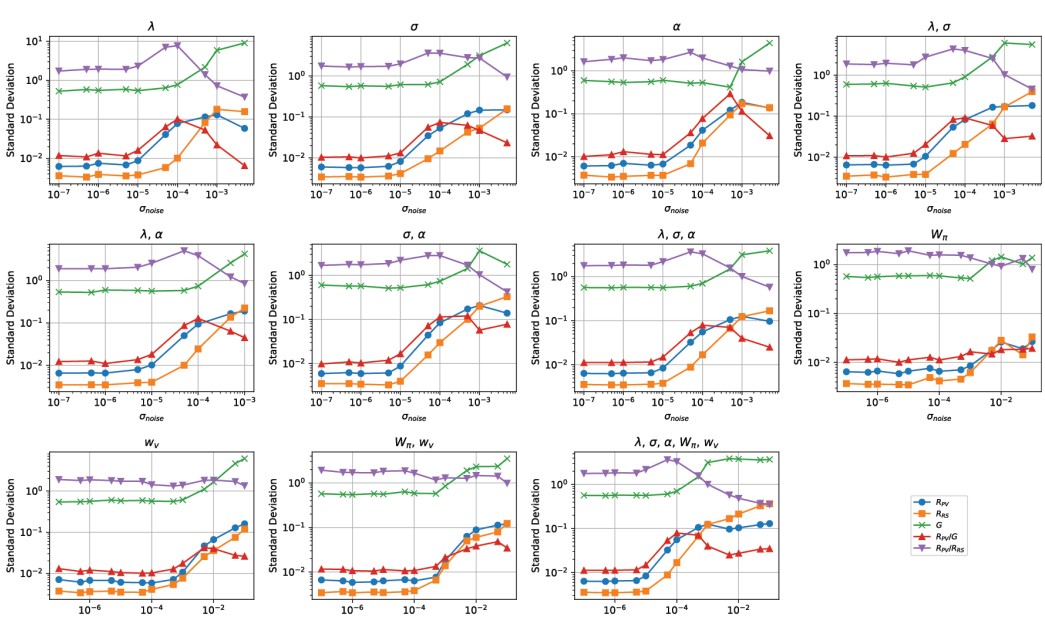

Supplementary Figure 7: **Noise amplitude monotonically influences population vector correlation and agent performance.** Adding Gaussian noise with increasing magnitude $[5x10^{-7}, 10^1]$ either in field parameters $(\alpha, \lambda, \sigma)$ or Actor-Critic $(W_\pi, w_v)$ influences the variance in Population Vector Correlation ($R_{PV}$, blue), Spatial Representation Similarity which is the dot product of field activity ($R_{RS}$, orange) and cumulative discounted reward ($G$, green). Low variance of $R_{PV}$ and $R_{RS}$ indicates high correlation as learning progresses. Low variance in $G$ indicates stable performance. When $G$ increases before decreasing as the noise amplitude increases, agent's navigation performance collapsed and the agent achieves 0 reward with low variance. A high ratio of variance in population vector correlation and reward maximization behavior ($R_{PV}/G$, red) indicates that there is an optimal noise amplitude which causes high variance in population vector correlation (low PV correlation) while demonstrating stable performance. A similar analysis can be performed using representational similarity ($R_{PV}/R_{RS}$, purple) to determine the optimal noise amplitude for high variance in population vector correlation but stable representation similarity as seen in Qin et al. (2023). Note that our agents are only optimizing for navigation behavior instead of representation similarity.

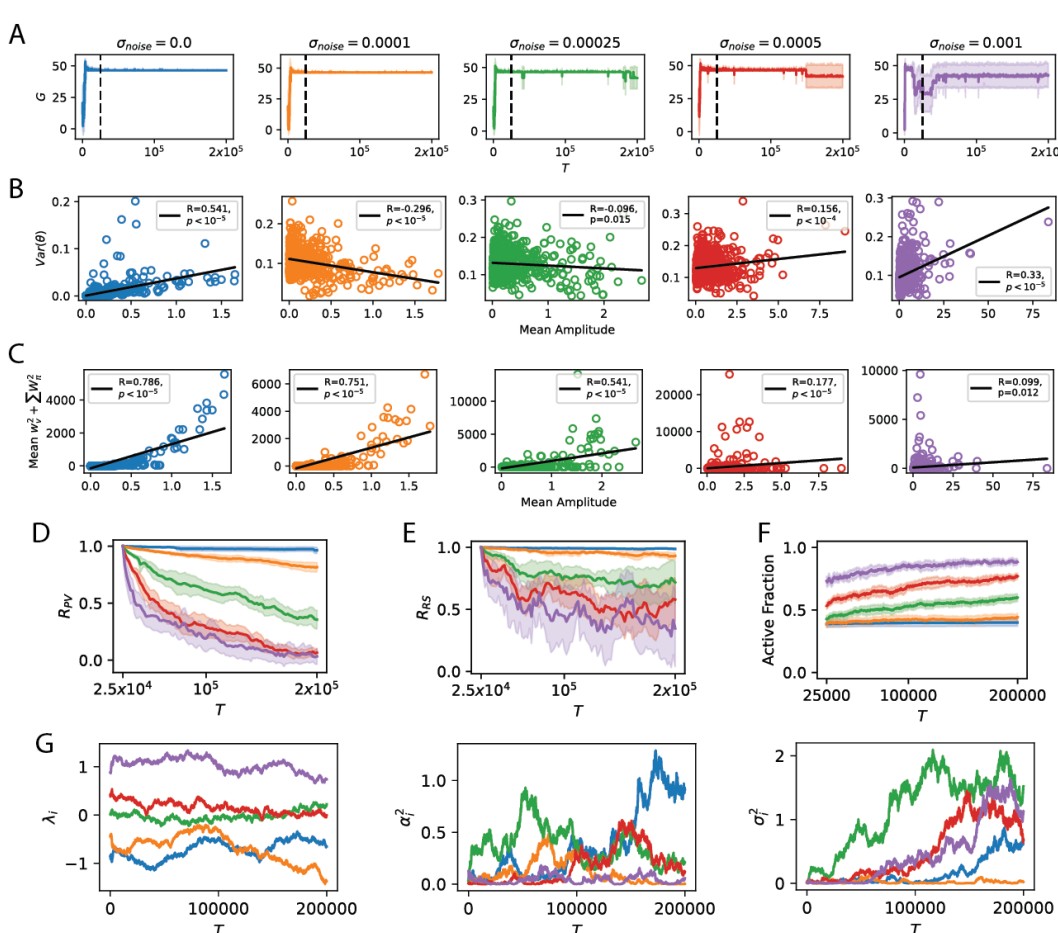

Supplementary Figure 8: **Influence of noisy fields on agent performance and field representation.** **(A)** Reward maximization performance variability increases when noise magnitude increases. **(B)** With no noise injection, variance in parameter update is initially positively correlated with field amplitude (blue). When a small amount of noise is added, fields with a larger mean amplitude show a smaller variance in change in parameter while fields with a smaller amplitude show higher variance. Conversely, when the magnitude of noise is further increased (purple), fields with a higher amplitude show higher variance in its parameters. **(C)** The correlation between mean amplitude and the magnitude of the readout weights (sum over all actions for squared actor weights and squared critic weights) is high and positively correlated when the noise magnitude is low. This correlation decreases and becomes weakly positive when $\sigma_{noise} = 0.001$. This supports the claim that in the low noise regime, fields with a high amplitude are more involved in policy learning and hence drift less or are more stable to maintain performance integrity. **(D)** Population vector correlation decreases at a faster rate than the similarity matrix when noise magnitude increases. **(E)** Representation similarity correlation decreases as the noise magnitude increases, but at a slower rate than PV correlation. **(F)** Proportion of fields that are active (average fraction of fields with firing rate less than 0.05, 0.1,0,25) continues to increase with higher noise magnitude. **(G)** Introducing Gaussian noise with zero mean and variance $N(0, 0.00025)$ to place field parameters during updates $\theta_{t+1} = \theta_t + \xi_t$ caused each place field's center, firing rate and width to fluctuate as trials progressed. See App. D for details. This causes each field's spatial selectivity to change over time. Specifically, each field's centroid ($\lambda$) shifted from its initialized location, firing rates fluctuated ($\alpha^2$) causing fields to gain or lose selectivity, and most fields increased in size ($\sigma^2$) while some did not. The first two were observed by Qin et al. (2023) who analyzed Gonzalez et al. (2019). Each color corresponds to the dynamics of a specific field, with 5 example fields shown.

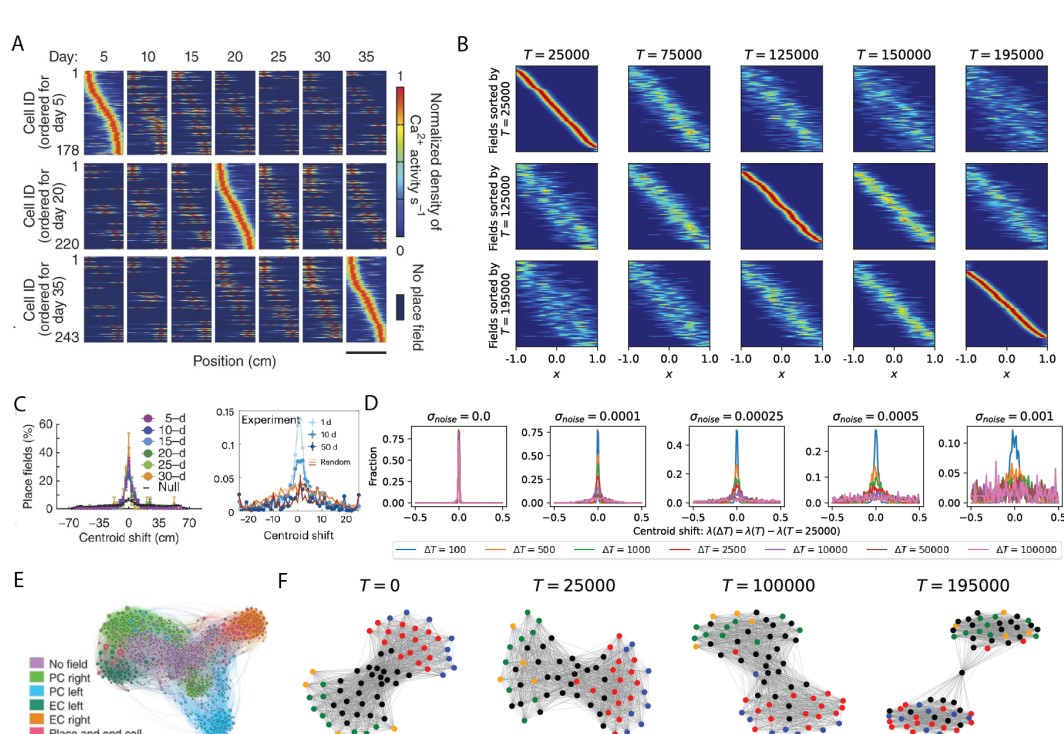

Supplementary Figure 9: **Noisy place field parameter update replicates drift dynamics seen in neural data.** **(A)** Place field spatial selectivity changes over days across four mice. Each place fields' centroid positions were sorted according to day 5, 20 and 35. Figure adapted from Fig. 3E-G, Ziv et al. (2013). **(B)** Place fields selectivity similarly changes across trials, after stable navigation performance was attained at trial 25,000. Each place field's centroid position was sorted according to trial 25,000, 125,000 and 195,000. As trials progress, spatial selectivity becomes distinctively different similar to Ziv et al. (2013) and Fig. 1G, de Snoo et al. (2023). **(C)** Probability distributions of centroid shifts along a 1D track at six (left, adapted from Fig. 3D, Ziv et al. (2013)) and three (right, adapted from Fig. 5H, Qin et al. (2023) who analyzed Gonzalez et al. (2019) data) different time intervals. Similar centroid shift away from the initialized position is also observed in Fig. 4B, Geva et al. (2023). **(D)** When no Gaussian noise is added to place field parameters $(\alpha, \lambda, \sigma)$, place field optimization alone does not cause centroids to shift as in neural data. Instead, adding small Gaussian noise ($\sigma_{noise} \in \{0.0001, 0.00025, 0.0005\}$) replicates the gradual shift in centroid position across trials (25,100 to 125,000). When the noise magnitude is high e.g. $\sigma_{noise} \geq 0.001$, centroids shift rapidly to a new location, similar to the random shuffle or null hypothesis used in Geva et al. (2023); Qin et al. (2023); Ziv et al. (2013). **(B-D)** Analysis was done for 64 place fields aggregated over 10 agents initialized with different seeds to have 640 fields in total. **(E)** Graph topology of place field activity in a 1D track show clustering of fields according to place encoding (PC) or end cell (EC, fields found at the end of track). Figure adapted from Fig. 4C, Gonzalez et al. (2019). **(F)** Example graph topology for one agent with $N = 64$ place fields with Gaussian noise $\sigma_{noise} = 0.00025$ added to field parameters. Each node indicates a place field's centroid position across learning, and the edge is weighted by the normalized (between 0 to 1) cosine distance between each node that is less than 0.55. Red, green, blue, orange, black nodes indicate centroids initialized at the reward, start, end of track near the reward, end of track near the start locations and the middle of the track respectively. As learning progressed, the cosine distance between each centroid changed and the ensemble representation rotated. Nevertheless, fields encoding the reward, start, and track were fairly stably as seen in Gonzalez et al. (2019), and the greater separation of clusters support the phenomenon where a high density of fields emerge at the reward and start locations.

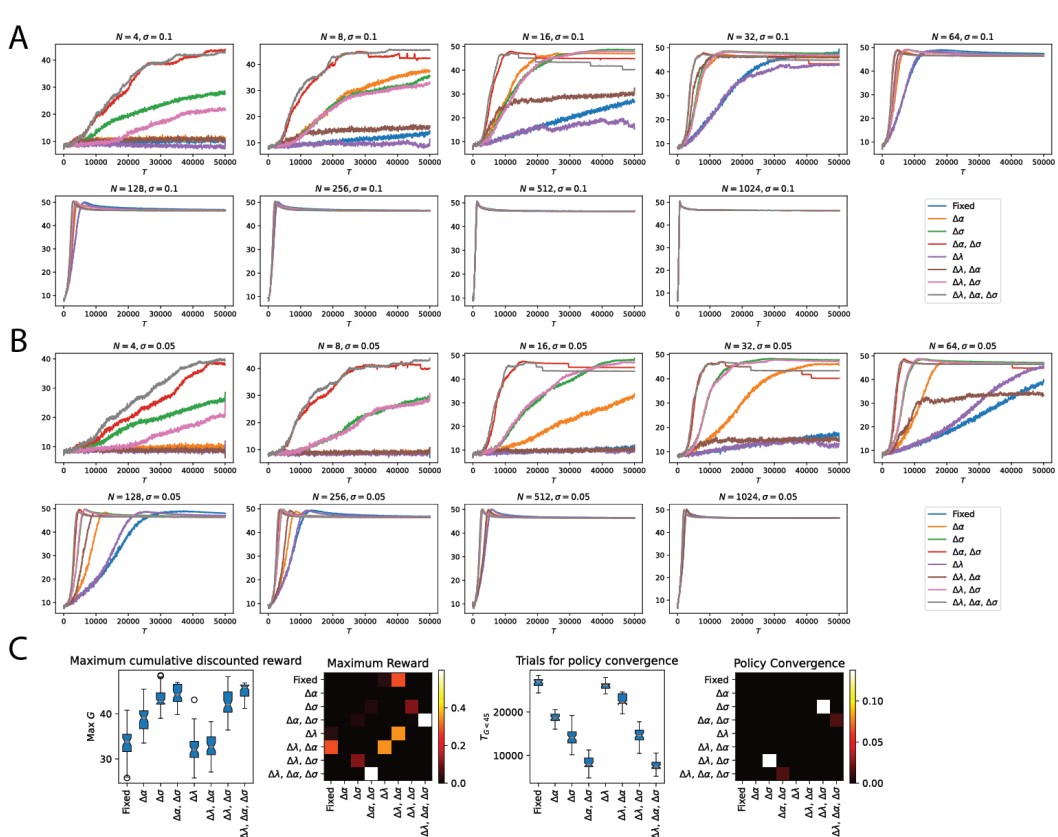

Supplementary Figure 10: **Influence of field width and number of fields on agent performance.** (A) Fields initialized with $\sigma = 0.1$ and (B) $\sigma = 0.05$. Policy learning is slower when initialized with a smaller field width. (C) Influence of field parameter optimization on the average maximum cumulative reward (left) and trial at which agent achieves cumulative discounted reward of 45 and above for the previous 300 trials (right). Correlation plot shows the p-value for a pairwise t-test performed to determine the influence of fields parameters on learning performance.

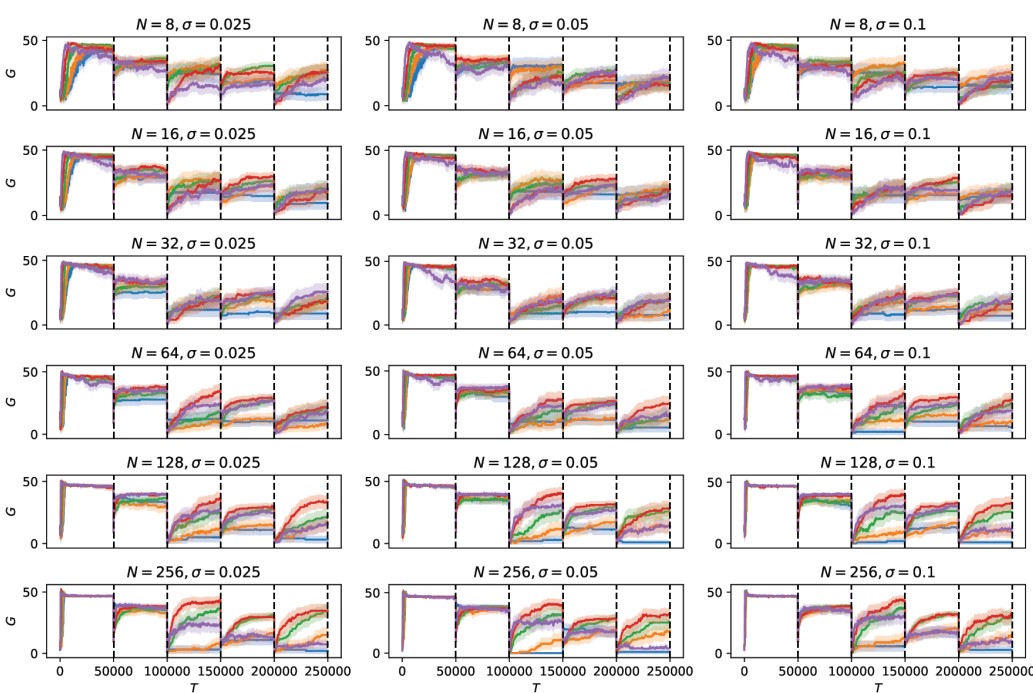

Supplementary Figure 11: **Influence of noise on new target learning performance in 1D track.** Increasing the number of place fields ($N$) and field widths ($\sigma$) led to a general increase in new target learning performance. When no noise was injected to field parameters ($\sigma_{noise} = 0.0$, blue), most agents struggled to learn to navigate to new targets and seem to be stuck in a local minima. Instead, noise magnitude of $\sigma_{noise} = 0.0005$ allowed agents to maximize rewards throughout the 250,000 trials. Increasing the noise magnitude beyond this ($\sigma_{noise} = 0.001$) negatively affected the agent's target learning performance, especially when the number of fields were low.

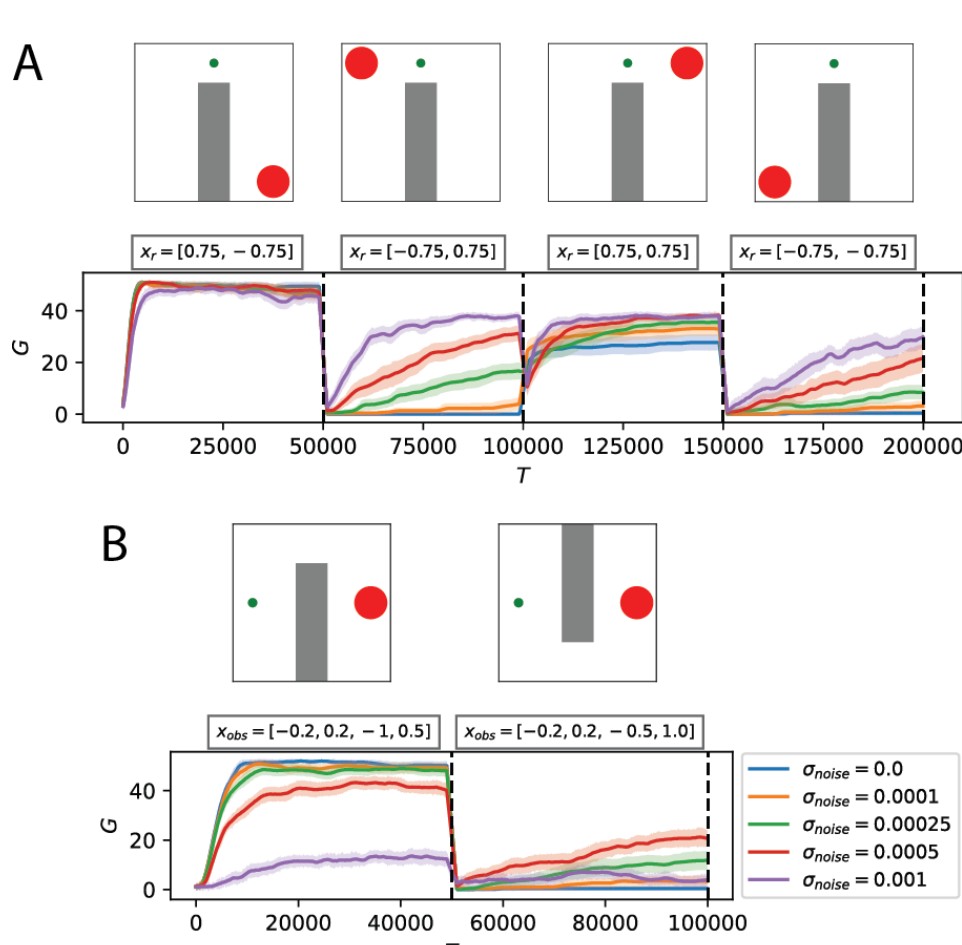

Supplementary Figure 12: **Influence of noise on learning performance in 2D arena with an obstacle.** **(A)** Agents started at the same location $x_{start} = (0.0, 0.75)$ and had to navigate to a target that changed to a new location every 50,000 trials following the sequence ($x_r \in [(0.75, -0.75), (-0.75, 0.75), (0.75, 0.75), (-0.75, -0.75)]$). Increasing the noise magnitude improved new target learning performance. **(B)** Agents learned to navigate to a target at $x_r = (0.75, 0.0)$ from a start location $x_{start} = (-0.75, 0.0)$ with an obstacle with coordinates ($x_{min} = -0.2, x_{max} = 0.2, y_{min} = -1.0, y_{max} = 0.5$) for the first 50,000 trials. After which, the location of the obstacle was shifted up to ($x_{min} = -0.2, x_{max} = 0.2, y_{min} = -0.5, y_{max} = 1.0$) while the start and target location was the same. Agents with a noise magnitude $\sigma_{noise} = 0.00025$ showed the highest average reward maximization performance followed by $\sigma_{noise} = 0.0005$. A high noise magnitude ($\sigma_{noise} = 0.001$) disrupted learning performance while agents without noisy field updates ($\sigma_{noise} = 0.0$) did not learn to navigate around the new obstacle. Note that field amplitudes and widths were clipped to be between $[10^{-5}, 2]$ and $[10^{-5}, 0.5]$ respectively to ensure the $\Sigma$ covariance matrix in 2D place fields remained valid for matrix inversion. Performance was averaged over agents initialized with different number of 2D place fields ($N \in \{64, 144, 256, 576\}$) with the diagonals of the field width initialized with $\Sigma = 0.01$ and constant amplitude $\alpha = 1.0$, over 30 different seeds. Shaded area is 95% CI.

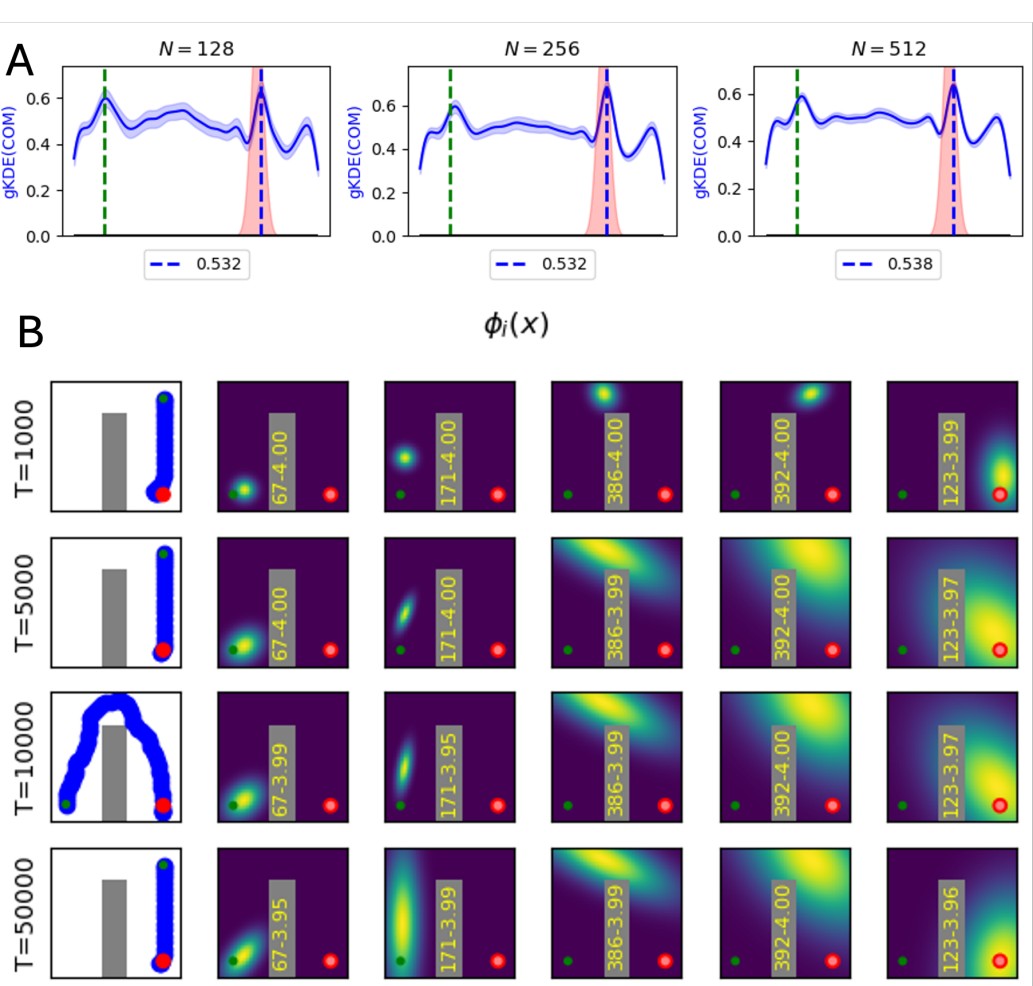

Supplementary Figure 13: **Using the same learning rates for the place field parameters and actor-critic recovers the same phenomena of a high field density emerging at reward location followed by the start location, and field elongation against the agent's trajectory.** (A) Each place field's amplitude, center and width were sampled from a uniform distribution of [0,1], [-1,1], [1e-5,0.1] respectively to model heterogeneous place field distribution. After learning, a high density (number) of fields emerged at the start (green dash) and reward (red area) location, similar to Fig. 1B (right) and Sup. Fig. 2B. This phenomenon is consistent across different numbers of place fields. Shaded area is 95% CI over 50 different seeds. (C) In a 2D arena with obstacles, place fields elongate from the reward location (red circle) back to the start location (green circle), while narrowing along the corridor with an obstacle (gray), similar to Fig. 2F. Learning rates for the actor, critic and place field parameters were $\eta = \eta_\theta = 0.0005$.

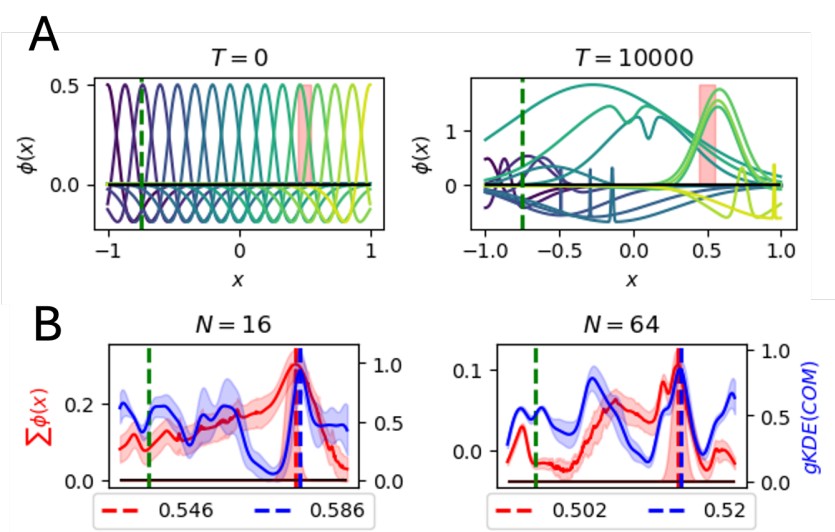

Supplementary Figure 14: **Center-surround place fields reproduces the emergence of a high density of fields at the reward location.** (A) Example of 16 center-surround fields uniformly distributed before (left) and after learning for 10,000 trials (right), with the learning rates for the center-surround place field parameters and policy network being the same ($\eta = \eta_\theta = 0.001$). Place fields near the reward shifted to the reward location while others elongated from the reward location back to the start location similar to Fig. 2C (bottom row). (B) A high field density (gKDE(COM)) and mean firing rate ($\sum \phi(x)$) emerged at the reward location for $N = 16$ (left) and $N = 64$ (right) when using center-surrounds fields. However, we do not see a high density emerging at the start location robustly. Further analysis is needed to verify if the representations learned by Gaussian basis functions and center-surround fields (difference of Gaussians) are similar, and if not why. Shaded area is 95% CI for 10 seeds.

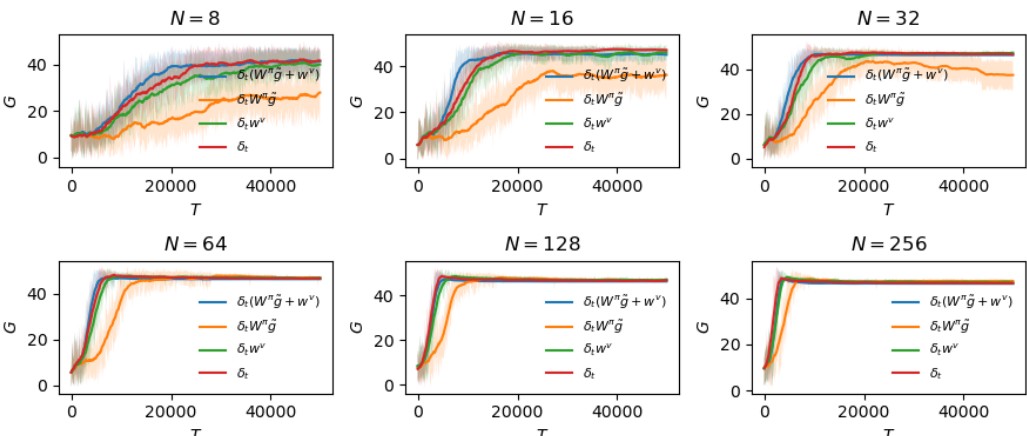

Supplementary Figure 15: **Difference in policy convergence when backpropagating temporal difference error through the actor and/or critic weights to optimize place field parameters.** We evaluated the speed of policy learning when optimizing place field parameters using (1) the actor weights $W^\pi$ multiplied by the normalized action vector $\tilde{g}_t = g_t - P$ and the critic weights $w^v$ (blue) (2) only the the actor weights multiplied by the normalized action vector (orange) (3) only the critic weights (green) (4) direct feedback of the TD error to modulate field parameters instead of backpropagating through the actor or critic weights, making it more biologiclly plausible (red). The combined objective used for place field parameter optimization achieved the fastest policy convergence when the number of fields was low ($N = \{8, 16, 32\}$) (blue). With more fields, using the critic weights (green) was almost as effective as the combined objective. Optimizing place field parameters using only the actor weights led to the slowest policy convergence (orange). Surprisingly, direct feedback of the TD error to modulate place field parameters shows the 2nd fastest policy convergence. Additional analysis is needed to determine the nature of representations learned by all four methods. Shaded area indicates 95%CI over 30 random seeds with place field amplitudes and widths uniformly initialized between $[0, 1]$ and $[10^{-5}, 0.1]$ respectively.

