# OpenReview forum: "A Model of Place Field Reorganization During Reward Maximization"
_ICLR.cc/2025/Conference — Submitted to ICLR 2025_

### Official Review · Reviewer_LQgu · 2024-11-02

**Soundness:** 3
**Presentation:** 3
**Contribution:** 3
**Rating:** 8
**Confidence:** 3

**Summary:**

This paper presents a computational model for hippocampal place field reorganization during reward-based navigation learning. The model integrates reinforcement learning via actor-critic with place fields represented through Gaussian radial basis functions. The learning system is capable to create high-density place fields near reward locations, field elongation along the trajectory, and drift in place fields. The TD signal is used to tune field properties such as amplitude, centre, and width. As in similar other studies, the use of noise helps learning and generalization.

**Strengths:**

- The paper is clearly written and well structured
- The paper makes an effort to provide a unified theory for multiple place field phenomena
- The integration of the actor-critic RL system with pace fields is biologically relevant and interesting
- The model is compared against the well known successor representation model
- The link between a normative approach via reward maximisation and the emergence of place field representation could be valuable to interpret biological dynamics.
- The use of noise to modulate the learning flexibility is interesting and potentially insightful for neuroscience
- The results are provided with 10 seeds runs and confidence intervals

**Weaknesses:**

- While the contributions are clear, it is less clear exactly in which why this model advances previous models and what we now understand better. I am also generally struggling with words such as 'recapitulate' that don't seem to mean much to me; do the authors means to say that this models reproduces for the first time the biological neural dynamics observed in the hippocampus? If so, it would be better to be more explicit.

- The authors mentioned very briefly about the use of backpropagation and the future requirement for a biologically plausible rule. I feel that this part is not sufficiently expanded. Can the author speculate whether the model would work at all with alternatives, e.g. STDP-like rules? And if so, how would that affect the model and the ability to reproduce biological neural dynamics?

- How foes noise in the model relate to biological systems? While there are a few relevant papers that are cited in that respect, it would useful to have a motivation that is more grounded in biological dynamics. E.g. it is known that neuromodulatory activity could be regulating exploratory or exploitative behavior, possibly affecting neural dynamics with different level of noise or signal to noise ratio.

- It's not clear to me how the model would work without rewards, as this is the case in biology where place cells and place fields develop also in the absence of reward. It would good to have at least some indications on how the model could be adapted to work without an explicit reward.

- Maybe I missed it, but the supplementary information says "Refer to the Github code repository for
implementation details." but I didn't find the link to it.

**Questions:**

The list of weaknesses contains questions. The authors are encouraged to consider them and possibly update the paper to account for this points and provide responses.

---

> ### Author Response · Authors · 2024-11-22
> **Simple model that reproduces several hippocampal dynamics using a singular objective**
>
> We would like to thank the reviewer for their insightful comments. We have added new Sup. Fig. 14 and uploaded the new manuscript with the clarifications and discussions based on your comments, which we will refer to henceforth. We will upload the code to run the place field agents as a .zip file in the supplementary.
>
> > While the contributions are clear, it is less clear exactly in which why this model advances previous models and what we now understand better. I am also generally struggling with words such as 'recapitulate' that don't seem to mean much to me; do the authors means to say that this models reproduces for the first time the biological neural dynamics observed in the hippocampus? If so, it would be better to be more explicit.
>
> Classical spatial navigation models are mechanistic bottom-up models where sub components like place, grid, head direction cell and dopamine activity are pieced together to reproduce the cellular dynamics (e.g. Successor Representation (Stachenfeld et al. 2017 Nature. Neuro,) or the navigation behavior (Kumar et al. 2024 arXiv 2106.03580). Alternatively, descriptive place field models can be easily manipulated to fit neural data (Maniali et al. 2024 arXiv), but do not demonstrate the emergence of new representations or reproduce behavior. These works explain the how and what questions respectively pertaining to place field dynamics. Normative models are top-down models that attempt to answer the why question, what is the global objective that causes the emergence of complex biological representations (e.g. Ganguli & Simoncelli 2014; Olshausen & Field 1996; Mante et al. 2013).
>
> To answer your question, yes, our model is the simplest that reproduces several place field neural dynamics observed in the hippocampus while capturing an animal's navigation learning behavior using the reinforcement learning framework. The three disparate dynamics can be reproduced when optimizing for a normative goal, without introducing too many constraints, which is intriguing. We believe this is an important contribution and we have clarified this contribution in the paper as requested. We will be happy to elaborate further.

---

> ### Author Response · Authors · 2024-11-22
> **Biological plausibility of model**
>
> > The authors mentioned very briefly about the use of backpropagation and the future requirement for a biologically plausible rule. I feel that this part is not sufficiently expanded. Can the author speculate whether the model would work at all with alternatives, e.g. STDP-like rules? And if so, how would that affect the model and the ability to reproduce biological neural dynamics?
>
> Backpropagation in the model was necessary to pass the temporal difference (TD) error through the actor and critic weights to get $\delta_t^{bp}$ in Eq. 48 before using this term to modulate the place field parameters in Eq. 49 to 51. This is the approach used in deep reinforcement learning literature (Wang et al. Nature Neuro.; Mnih et al. 2015 Nature), but it is considered to be biologically implausible.
>
> Conversely, several mechanistic models such as those that use Hebbian plasticity, STDP or BTSP have been shown to replicate some, not all of the phenomena. Based on our current model that uses reward-like signal, we think that we might need a reward-modulated or error-modulated learning rule e.g. Temporal Difference error modulated STDP (Freamux et al. 2013 Plos Comp Bio.) or TD Hebbian Learning (Kumar et al. 2022 Cerebral Cortex) to replicate all the phenomena. Previous biologically plausible algorithms that perform feature learning (e.g. Miconi et al. 2017; Murray 2019) have not looked at using the TD error for representation learning yet. This is the idea we hope to pursue as future work.
>
> Additionally, we have preliminary results that show that Direct Feedback of the TD error to the place field parameters might be sufficient to get similar place field dynamics and learning performance (Sup. Fig. 14). This means that we do not need to perform backpropagation and instead directly use the TD error to modulate the place field parameters (to use $delta_t$ instead of $delta_t^{bp}$ for Eq. 49 to 51). This is exciting as it is biologically plausible and fits the neuroanatomy seen in Russo & Nestler 2013 Nature Rev. Neuro where there are strong dopaminergic projections from the Ventral Tegmental Area to the hippocampus. However, we need to do additional analysis and improve on our current theoretical analysis for why this works.
>
> Nevertheless, we have elaborated on this point in the discussion section and included Sup. Fig. 14. for your perusal.

---

> ### Author Response · Authors · 2024-11-22
> **Biological origin of noise to induce drift**
>
> > How foes noise in the model relate to biological systems? While there are a few relevant papers that are cited in that respect, it would useful to have a motivation that is more grounded in biological dynamics. E.g. it is known that neuromodulatory activity could be regulating exploratory or exploitative behavior, possibly affecting neural dynamics with different level of noise or signal to noise ratio.
>
> Synaptic plasticity is noisy due to several molecular mechanisms (change in NMDA receptors, ion channels, etc.) and pathways (Allen & Stevens 1994; Mongillo, Rumpel, Loewenstein 2017; Kappel, Habenschuss, Legenstein, Maass 2015; Attardo, Fitzgerald, Schnitzer 2015). Extrinsic factors such as glial (Sancho, Contreras, Allen 2001) and astrocytes (Bernardinelli, Muller, Nikonenko 2014; Pitta, brunel, Volterra 2016) modulate synaptic plasticity as well. These experimental results suggest that it is reasonable to add small Gaussian noise to synapses or neural parameters during learning.
>
> Additionally, Krishnan et al. 2022 Nature Comms. causally showed that dopamine responses directly influenced place field remapping. In a followup paper Krishnan et al. 2023 bioRxiv showed that CA1 place fields drifted less when the reward expectancy was high (probability of being rewarded) and place fields drifted more when the reward expectancy decreased (unrewarded condition). This suggests that dopamine responses could be causally manipulating the drift rates in the hippocampus. Similarly, in Fig. 4 and Sup. Fig. 7, we showed that different noise levels non-monotonically influenced new target learning performance.
>
> Another origin of noise could be coming from the entorhinal cortex, either due to unstable attractor manifold (as suggested by Reviewer J18N) or noisy synaptic plasticity between presynaptic grid cells and postsynaptic place cells (Qin et al. 2023 Nature Neuro.). It is definitely an appealing hypothesis that an underlying attractor mechanism, when stable, reduces place field drift rates, and when unstable, contributes to a higher drift rate. This mechanism coupled with dopamine responses could modulate place field drift rates. This will be an interesting hypothesis to test in a computational model and verify in experiments.
>
> We have added the biological origin of noise in place fields in the discussion section. Thank you for this suggestion.

---

> ### Author Response · Authors · 2024-11-22
> **The need to explore both reward and non-reward based objectives**
>
> > It's not clear to me how the model would work without rewards, as this is the case in biology where place cells and place fields develop also in the absence of reward. It would good to have at least some indications on how the model could be adapted to work without an explicit reward.
>
> We agree that place fields do reorganize in the absence of rewards, as a form of latent learning (Tolman, 1948). However, in our current model, when rewards are absent, place fields will not show much reorganization since the objective that the place fields are optimizing for are reward dependent. We started with a reward based objective as most theory in reinforcement learning is centered around rewards (Sutton & Barto 2018; Kakade 2023; Agarwal et al. 2021; Bordelon et al. 2023 NeurIPS), and we were curious if a reward-like learning signal could replicate disparate neural phenomena.
>
> Nevertheless, we believe the current model can be easily extended to optimize for non-reward based objectives since it performs gradient computation, and theoretically study (E.g. Eq. 7 and Eq. 65) how these objectives influence place field reorganization in a follow up paper. As a starting point, we will explore the objectives used in Fang 2024 ICLR which is to predict the next state given the current state, and Foster et al. 2000 Hippocampus which is to predict a low-dimensional metric representation from place field activity. We have included our model’s reward dependency as a limitation in the discussion section and have described non-reward based objectives as future work. Thank you for highlighting this.

---

> ### Author Response · Authors · 2024-11-22
> **Code for implementation**
>
> > Maybe I missed it, but the supplementary information says "Refer to the Github code repository for implementation details." but I didn't find the link to it.
>
> We included the github code in the original draft but removed the link to maintain anonymity. We have uploaded a minimal python code to train agents in a 1D track and 2D arena with an obstacle as a supplementary .zip file as requested. Thank you for your interest!

---

> > ### Comment · Reviewer_LQgu · 2024-11-27
> >
> > I thank the authors for carefully answering my questions in the weakness section. I consider the replies as well thought and helpful to improve the quality of the paper if well integrated in the paper draft. In light of these, I maintain my overall positive assessment.

---

> ### Author Response · Authors · 2024-11-28
> **Thank you for the feedback and support!**
>
> We would like to thank the reviewer for their insightful questions and for continued support! The manuscript is in a much better shape from where it started. We look forward to continuing discussions!

---

### Official Review · Reviewer_J18N · 2024-11-03

**Soundness:** 3
**Presentation:** 4
**Contribution:** 3
**Rating:** 6
**Confidence:** 4

**Summary:**

This paper develops a reinforcement learning model that uses tunable place fields to simulate learning dynamics observed in rodent navigation. The model represents place fields as Gaussian radial basis functions connected to an actor-critic RL framework, updating field parameters to maximize accumulated rewards based on temporal difference error. The model replicates key experimental findings: increased place field density near rewards, field expansion along navigational trajectories, and field drift that doesn’t disrupt performance. The model suggests a potential framework for studying place fields learning dynamics, offering predictions for future hippocampal experiments.

**Strengths:**

Explaining the emergence of three observations about place field reorganization in a normative approach.

Comparing with competitive model, such as successor representation.

Ablation study to understand contribution of different place field parameters to policy learning.

Well written, well organized.

**Weaknesses:**

The learning rate for place field parameters are 100 times slower than the actor and critic weight parameters. This separation of learning process by implementing different learning rates is not biologically grounded. Are there any potential biological mechanisms that might support different timescales of plasticity.

The pace fields has been found to have more complex shapes, rather than simple gaussian shape. For example, the center-surround structure in the place field (which could be modeled as difference of gaussians) was shown to have important functional role (Sorscher et al 2022 Neuron). It is not clear how using more complex place field shapes might affect the model's results or conclusions.

The number of place fields appears to be a crucial hyperparameter, as the model’s resemblance to experimental data is observed only within a specific range of values for this parameter. The authors examined the effect of the number of place fields on model performance in Supplemental Figures 6 and 7. However, it is important to explore methods to enhance the model’s robustness to variations in this parameter, which is relevant to the model's potential scalability to larger environmental spaces.

Randomness in field parameters, rather than in actor/critic weights, appears essential for the RNN to replicate field drifting and perform tasks with changing targets. This distinction highlights two types of noise in the RNN, each with distinct functional roles; however, a discussion on the possible biological origins of this noise and its functional distinctions is lacking. In the current work, field drifting is modeled by manually injecting noise into field parameters. Could the ring attractor mechanism identified in Sorscher et al. (2022, Neuron) provide a more mechanistic interpretation of field drifting?

**Questions:**

The paper takes a normal approach, using gaussian radial basis function to model place fields. There are multiple works that modeling place field using RNN without specific place field parameter assumptions, but taking them as emergent features. Could the place field reorganization be explained in such approach?

Could the authors consider incorporating center-surround structures or other non-Gaussian shapes and discuss how this might impact their findings?

The learning rate for place field parameters and the actor/critic weight parameters possess distinct time scales.  Could the authors explore how varying the learning rate ratio affects the model's behavior and ability to replicate experimental findings. Could the model still work after removing this assumption?

---

> ### Author Response · Authors · 2024-11-22
> **Same learning rate shows consistent reorganization phenomena**
>
> We would like to thank the reviewer for their insightful comments. We have added new Sup. Fig. 4, 12, 13, and uploaded the new manuscript with the recommended clarifications, discussions and references based on your comments, which we will refer to henceforth.
>
> > The learning rate for place field parameters are 100 times slower than the actor and critic weight parameters. This separation of learning process by implementing different learning rates is not biologically grounded. Are there any potential biological mechanisms that might support different timescales of plasticity.
>
> > The learning rate for place field parameters and the actor/critic weight parameters possess distinct time scales. Could the authors explore how varying the learning rate ratio affects the model's behavior and ability to replicate experimental findings. Could the model still work after removing this assumption?
>
> Separation of timescales has been occasionally observed across various brain regions e.g. CA3 exhibits slower dynamics compared to the CA1 region (Dong et al., 2021, Nature Comms.). Similarly, differing rates of dynamics have been reported between the striatum and hippocampus (Segar & Cincotta, 2006). However, it is important to emphasize that a separation of timescales is not required for our place field model to replicate the observed reorganization behavior (as shown in Sup. Fig. 12). This is because we compute the gradients to optimize both place field parameters and the actor-critic components, allowing us to use the same learning rates for both layers, similar to the approach used in deep reinforcement learning (Wang et al., 2018, Nature Neuro.; Mnih et al., 2015, Nature). The separation of timescales was an important assumption for our perturbative approximation and theoretical analysis (see Eq. 7 and Eq. 65). However, it is not a prerequisite for the model's function.
>
> Supplementary Figure 12 illustrates that even when the same learning rate ($\eta=\eta_\theta=0.0005$) is applied to both the actor-critic and place field parameters, learning is stable. The model shows that a high density of fields emerges at both the reward and start locations on a 1D track, and also exhibit field elongation against the agent's trajectory in a 2D arena. Therefore, it is not essential for our model’s performance and place field representation learning. We have clarified this relaxation of learning rates in the revised manuscript, highlighting that such separation is not necessary to observe the same phenomena. Supplementary Figure 12 serves as empirical support.

---

> ### Author Response · Authors · 2024-11-22
> **Center-surround fields reproduce high density at reward location**
>
> > The pace fields has been found to have more complex shapes, rather than simple gaussian shape. For example, the center-surround structure in the place field (which could be modeled as difference of gaussians) was shown to have important functional role (Sorscher et al 2022 Neuron). It is not clear how using more complex place field shapes might affect the model's results or conclusions.
>
> Yes, place fields do exhibit complex shapes and there have been several place field descriptive models that capture this heterogeneity (Mainali 2024 bioRxiv; Luo et al. 2024 eLife; Sorscher et al. 2022 Neuron). Thank you for the reference, we have added these models to our discussion section and we will explore them as future work.
>
> > Could the authors consider incorporating center-surround structures or other non-Gaussian shapes and discuss how this might impact their findings?
>
> Based on your suggestions, Sup. Fig. 13 shows a simulation with a 10 different seeds using the center-surround place fields (using a difference of gaussians). Similar to the gaussian basis functions, using the center-surround features caused a high density of fields to emerge at the reward location with other fields elongating backwards from the reward location to the start location. However, we do not robustly see a high density emerging at the start location. We need to perform additional experiments to quantify if center-surround will lead to a significant difference in representations learned. Nevertheless, Sup. Fig. 13 shows that the model still works to show the same reorganization behavior when the same learning rate is used for the actor-critic and the place field parameters.

---

> ### Author Response · Authors · 2024-11-22
> **Biological origins of noise to induce drift**
>
> > Randomness in field parameters, rather than in actor/critic weights, appears essential for the RNN to replicate field drifting and perform tasks with changing targets. This distinction highlights two types of noise in the RNN, each with distinct functional roles; however, a discussion on the possible biological origins of this noise and its functional distinctions is lacking. In the current work, field drifting is modeled by manually injecting noise into field parameters. Could the ring attractor mechanism identified in Sorscher et al. (2022, Neuron) provide a more mechanistic interpretation of field drifting?
>
> Could we clarify that by the RNN, you are referring to our model that uses gaussian basis functions?
>
> Synaptic plasticity is noisy due to several molecular mechanisms (change in NMDA receptors, ion channels, etc.) and pathways (Allen & Stevens 1994; Mongillo, Rumpel, Loewenstein 2017; Kappel, Habenschuss, Legenstein, Maass 2015; Attardo, Fitzgerald, Schnitzer 2015). Extrinsic factors such as glia (Sancho, Contreras, Allen 2001) and astrocytes (Bernardinelli, Muller, Nikonenko 2014; Pitta, Brunel, Volterra 2016) modulate synaptic plasticity as well. These experimental works suggest that it is reasonable to add small Gaussian noise to synapses or neural parameters during learning.
>
> Additionally, Krishnan et al. 2022 Nature Comms. casually showed that dopamine responses directly influenced place field remapping. In a followup paper Krishnan et al. 2023 bioRxiv showed that CA1 place fields drifted less when the reward expectancy was high (probability of being rewarded) and place fields drifted more when the reward expectancy decreased (unrewarded condition). This suggests that dopamine responses could be casually manipulating the drift rates in the hippocampus.
>
> The ring attractor mechanism for grid fields in the entorhinal cortex, could definitely be a possible mechanism to control noise and drift rates in the hippocampus. Qin et al. 2023 Nature Neuro. demonstrated that noise added to synapses from grid cells to place cells induced representational drift. It is definitely an appealing hypothesis that an underlying attractor mechanism, when stable exhibits less noisy firing rates, reducing place field drift rates, and when unstable is more noisy, and hence contributes to a higher drift rate. This mechanism coupled with dopamine responses could modulate place field drift rates. This will be an interesting hypothesis to test in a computational model.
>
> If the Basal Ganglia or striatum, modeled as the actor-critic, is a readout of hippocampal representations to form sensory-response associations, it should be less prone to noisy computations to achieve stable behavior. The difference in the magnitude of noise in the hippocampus and the striatum needs to be experimentally verified.
>
> We have added the biological origins of noise in place fields and your reference in the discussion section. Thank you for this suggestion.

---

> ### Author Response · Authors · 2024-11-22
> **Weak to rich feature learning regime**
>
> > The number of place fields appears to be a crucial hyperparameter, as the model’s resemblance to experimental data is observed only within a specific range of values for this parameter. The authors examined the effect of the number of place fields on model performance in Supplemental Figures 6 and 7. However, it is important to explore methods to enhance the model’s robustness to variations in this parameter, which is relevant to the model's potential scalability to larger environmental spaces.
>
> This is a good question. How robust is the model to different number of place fields for flexible computation to navigate in both small and large environments?
>
> Sup. Fig. 4 shows that as we increase the number of place fields, the change in each field’s parameter decreases, indicating that our current model is in a weak feature learning regime for high N. Instead, we need to initialize our model such that it will be in a rich feature learning regime where we scale the readout weights and learning rates accordingly (Chizat et al. 2019 NeurIPS, Yang and Hu 2021 ICML). A model in a rich feature learning regime might be able to show robustness to the parameter N in larger environments such as to model Lee et al. 2020 Cell who had a 42m track and perhaps extend Eliav et al. 2021 who had a 300m track.
>
> Thank you for this suggestion, this is definitely a relevant idea to explore using a spatial navigation model and for place cell research. We will conduct experiments to include in this paper or as a follow up, on the difference in place field reorganization between the lazy and rich regimes.

---

> ### Author Response · Authors · 2024-11-22
> **Generality and flexibility of model to incorporate other place field descriptions**
>
> > The paper takes a normal approach, using gaussian radial basis function to model place fields. There are multiple works that modeling place field using RNN without specific place field parameter assumptions, but taking them as emergent features. Could the place field reorganization be explained in such approach?
>
> The main benefit of using Gaussian features is its simplicity for theoretical analysis. It would definitely be possible to perform the same simulations using other descriptions of place fields e.g. feedforward (Maniali et al. 2024 bioRxiv; Kumar et al. 2022 Cerebral Cortex) or RNN, though it could be more difficult to interpret the learning dynamics. We have indicated that this will be our future work in the discussion section. Nevertheless, we believe that we should see the same phenomena, specifically for a high field density emerging at the reward location (e.g. Fang et al. 2024 ICLR) and noisy features improving flexible learning (e.g. Dohare et al Nature 2024).

---

> > ### Comment · Reviewer_J18N · 2024-11-26
> >
> > The authors have address my questions and concerns; therefore,  I am increasing my score to 6.

---

> ### Author Response · Authors · 2024-11-28
> **Thank you for the feedback**
>
> We would like to thank the reviewer for their insightful feedback and for raising the score! The manuscript is in a much better shape from where it started. We look forward to continued discussions.

---

### Official Review · Reviewer_ifiL · 2024-11-09

**Soundness:** 3
**Presentation:** 2
**Contribution:** 1
**Rating:** 3
**Confidence:** 5

**Summary:**

This paper models the reorganisation of place fields on linear track by backpropagating the reward-based TD error gradient to the parameters of Gaussian place field basis features. The methods are sound and paper is presented well, however I have reservations about the conclusions drawn, particularly the links to changes in field density, neural drift and the reliance on a reward signal for the learning - which are at odds with certain aspects of the experimental hippocampal literature which the model is trying to explain

**Strengths:**

The paper provides and executes a novel idea with the intention of explaining certain aspects of the hippocampal literature. The observed backwards skewing of place fields, opposite to the direction of motion, provides an alternative account of place field skews (Mehta et al 2000), which so far has mainly been explained by predictive mechanisms (e.g. Mehta et al 2000) like the successor representation (Stachenfeld et al 2017).

**Weaknesses:**

The model claims to find increased place field density at reward locations - however if the definition of field density from the experimental literature was used, the authors would find their the model actually yields the opposite results to what is observed in the experimental literature. To be specific, Gauthier and Tank 2018 find increased density of field centre of masses - i.e. fields are more numerous at goal locations, however what the authors show is mean activity of place firing rate is greater at at the goal location, which is caused by a sparse number of place fields at the goal location with incredibly high firing rate (in order to create a ramping value toward the goal location). In fact, the majority of the place field centre of masses are shown to expand and skew backwards down the track, opposite to the direction of motion (as reported in the next result) thus if the authors used the density of field centre of masses - as the experimental literature does - they would actually find an increased field density in the locations on the track preceding the goal location, and a decreased field density at the goal.

The authors also claim the model captures observed neural drift of hippocampal place cells - this is quite a stretch. Place cells have been shown to be stable for very long periods of time - up to 153 days (Thompson & Best 1990). Indeed, reward expectation has actually been shown to decrease neural drift, opposite to the claims of the authors here (Krishnan & Sheffield 2023). Essentially what the the authors show is drift in the population vector over time - which is observed in the hippocampus - however the authors' drift is because place fields actually drift around the track, which is not observed in real life - place fields are generally stable. In fact, it is likely hippocampal neurogenesis, which is heavily triggered by exercise, is responsible for driving some features of the observed representational drift (Snoo et al 2023).

Finally the comparison to successor representation (SR) is apt, however a favourable quality of the SR model is it is not dependent on an external reward signal to trigger these changes, and thus can explain phenomena such as latent learning when no reward is present. In fact the model presented here implies experimentalists would observe huge differences in hippocampus based on whether or not reward is present, when if fact periods of exploration in the absence of reward only improves future reward based learning (Tolman 1948)

**Questions:**

Based on the above, I would recommend the authors:
1. Use the same measures on their data that the experimental results they are attempting to explain use
2. Consider dropping the neural drift part of the results, as the underlaying mechanism in the model is at odds with many aspects of the hippocampal literature
3. Consider focussing on a set of experimental results better suited to the model e.g. the Dupret et al 2010 cheeseboard task

---

> ### Author Response · Authors · 2024-11-22
> **Model reproduces high density (number) of place fields at reward location**
>
> We would like to thank the reviewer for their insightful comments. We have rectified Fig. 1, Sup. Fig. 1,8, added new Sup. Fig. 2, and uploaded the new manuscript with the recommended references based on your comments, which we will refer to henceforth.
>
> > The model claims to find increased place field density at reward locations - however if the definition of field density from the experimental literature was used, the authors would find their the model actually yields the opposite results to what is observed in the experimental literature.
>
> We agree that density is described as the number of fields in a location (Gauthier et al. 2018; Lee et al. 2020). We defined density as the mean firing activity at a location instead of the number of fields (based on the center of mass) as a higher firing rate (amplitude) at a location influences policy learning to a greater extent (Sup Fig. 8C) than a place field’s center of mass or a single field’s firing rate alone. That is, a small number of place fields with large firing rates is more influential than a large number of fields with extremely small firing rates. Furthermore, optimizing for a field’s center of mass is not as beneficial in policy learning as compared to a field’s width and amplitude. Instead, optimizing for a field’s center destabilizes policy learning (Fig. 4A). Hence, we wanted to motivate a more holistic definition of density for policy learning.
>
> Nevertheless, we agree that we should have used the canonical density definition (number of fields in a location) instead. Fig. 1 and Sup. Fig. 1 in the paper have been re-plotted with the density defined as the number of fields with their center of mass in a location as in the literature. The blue line in the new figures show the number of place fields in a location measured using their Center of Mass (COM) and smoothed using a Gaussian kernel density estimate d(x) = gKDE(COM). Based on the canonical density definition, we do concede that the example distribution shown in the original Fig. 1B shows a lower density at the reward location, as pointed out. However, we would like to argue that this is not a good example and instead, a high density of fields do in fact emerge at the reward location robustly as we observe in data.
>
> In the newly attached Fig. 1B,C and Sup. Fig. 1, we show how optimizing different combinations of place field parameters (center $\lambda$, amplitude $\alpha$,width $\sigma$) influences place field reorganization. When we optimize a low number of fields e.g. N=16 that have been uniformly distributed along the 1D track with the same width and amplitude, a higher number of fields with small firing rates organize at the home location (Sup. Fig. 1A), contrary to neural data. However, when we increase the number of tunable fields that are uniformly distributed along the 1D track with the same width and amplitude e.g. N=256, a high number of fields do organize at the reward location (Sup. Fig.1B), similar to neural data.
>
> More importantly, we randomly sampled each place field’s amplitude, center and width parameters from a uniform distribution between [0,1], [-1,1] and [1e-5, 0.1] respectively and optimized the parameters using the same algorithm over 50 different seeds. The new Fig. 1B and Sup. Fig. 2A,B and. shows that a high density (number) of fields do emerge at the reward location followed by the home location robustly across the 50 seeds when we optimize 4 to 512 place fields.

---

> ### Author Response · Authors · 2024-11-22
> **Theory predicts fields move to the reward and start location**
>
> Additionally, we would like to refer back to Eq. 7 and Appendix B that derives a perturbative approximation of our place field model. A theoretical analysis of the gradient flow shows that place field centers will straddle between a frequently visited location \bar \lambda i.e. reward location and the start location \mu_x, depending on its relative position in the track.
>
> Fields will shift depending on the value of the location (indicated by the magnitude in $w_{v,i}$). In the beginning of learning, the reward location will have a non-zero, positive value while the home location tends to have a value close to zero. Hence, fields near the reward location will move first. In the later stages of learning, when the home location increases in value, due to credit assignment by the TD error, fields near the home location will shift.
>
> The direction of shift depends on the location in which the field was initialized at. When a field is initialized near the reward location ($\bar \lambda \approx \lambda$ ) , it will have a tendency to move towards the reward so that the difference $\bar \lambda - \lambda$ goes towards zero causing $\Delta \lambda(t)$ in Eq. 66 to go towards zero. However, $\mu_x - \lambda$ will be nonzero, causing a pull on that field to move closer to the home location $\mu_x$. Similarly, when a field is initialized near the home location ($\mu_x \approx \lambda$), it will have the tendency to move towards the home location while having a pull towards the reward location. Hence, each field will continue to dynamically change its location until a stable distribution is achieved that straddles between the reward location and home location. Based on simulations, when there is a low number of fields, the gradient optimization process tries to balance field centers between the reward and start location, though an equilibrium point might not be achievable. When the number of fields is high, an equilibrium point can be attained.
>
> This equation also explains why we see a backward shift in the field center towards the home location in later stages of learning. We hope that using the canonical density metric, new simulations and theoretical analysis in the paper will be sufficient to argue that our model does show a higher number of fields organizing both at the reward and home locations as seen in neural data. We look forward to the reviewer’s perspective on these.

---

> ### Author Response · Authors · 2024-11-22
> **New data shows place fields do drift around the track**
>
> > ... however the authors' drift is because place fields actually drift around the track, which is not observed in real life - place fields are generally stable.
>
> – The critique that place fields are generally stable and do not drift around the track is not entirely true. After learning to navigate, an individual place field’s spatial selectivity changes over the course of days such that their population vector correlation decreases (Ziv et al. 2013 Nature Neuro.; Gonzalez et al. 2019 Science). Place cells do not only lose or gain place fields over the days (due to neurogenesis (Snoo et al. 2023) or other mechanisms such as BTSP), but in a recent analysis, place field centroids also shift or move along the track such that their centroid shifts from their initialized position to a new position over a matter of days.. This drifting of fields along the 1D track is seen in Figure 5 from Qin et al. 2023 Nature Neuro. where they analyzed place field activity from Gonzalez et al. 2019 Science.
>
> While we do not explicitly model neurogenesis here, by introducing Gaussian noise with zero mean and different magnitudes of variance, our place field model captures (1) individual place fields moving away from their initialized position (Sup. Fig. 8G left, change in $\lambda_i$) (2) each place field’s firing rate fluctuates such that they lose or gain the place field (Sup. Fig. 8G center, change in $\alpha_i^2$) and (3) individual field’s size fluctuates while most of them increases with learning (Sup. Fig. 8G right, change in $\sigma_i^2$). (1) and (2) has been observed in neural data as shown by Qin et al. 2023. Perhaps (2) also captures similar dynamics to neurogenesis? We have included this reference as a possible drift mechanism in the paper.
>
> Furthermore, Krishnan & Sheffield 2023 showed that place fields drifted more in the absence of reward (low reward expectancy) and drifted less in the presence (high reward expectancy). In our current model, we did not modulate the speed of drift using reward expectancy although reward expectancy dependent variance is something we hope to model next.
>
> We hope these would clarify the reviewer’s concerns about drift. We will be more than happy to provide additional justifications in terms of neural data.

---

> ### Author Response · Authors · 2024-11-22
> **The need to explore both reward and non-reward dependent objectives**
>
> > a favourable quality of the SR model is it is not dependent on an external reward signal to trigger these changes, and thus can explain phenomena such as latent learning when no reward is present. In fact the model presented here implies experimentalists would observe huge differences in hippocampus based on whether or not reward is present,
>
> You are correct to point out that place fields do reorganize in non-rewarded settings as studied using the Successor Representation and latent learning contexts. In this paper, we wanted to explore how a normative reward maximization objective influences place field reorganization for policy learning, a question that has not been answered prior. Furthermore, the reward setting has been well studied in reinforcement learning with some theoretical underpinning (Sutton & Barto 2018; Kakade 2023; Agarwarl et al. 2021; Bordelon et al. 2023 Neurips) compared to the non-reward setting.
>
> Referring back to Mehta et al. 1997 PNAS, rats were pretrained using food reinforcement (reward) to run in a triangular and rectangular track, before recording during the 17 non-rewarding laps. Hence, rats could have a reward expectancy or prediction error while running for 17 laps. Although rewards were not explicitly given during the 17 laps, place field reorganization seen in Mehta et al. 1997 could also be driven by a residual reward expectation signal during pretraining, or there could have been a self-defined reward-like signal e..g information maximization such as novelty or curiosity. Hence, we were curious if a reward based objective could first replicate a non-rewarding phenomenon, and to our surprise, it did.
>
> Nevertheless, we believe the current model can be easily extended to optimize a non-reward based objective and theoretically study (e.g. Eq. 7 and Eq. 65 in Appendix B) how these objectives can influence place field reorganization in a follow up paper e.g. using the non-reward dependent objectives seen in Fang et al. 2024 ICLR which is to predict the next state given the current state and Foster et al. 2000 Hippocampus which is to predict a low-dimensional metric representation using place field activity.
>
> We have included our model’s reward dependency as a limitation in the discussion section and have described non-reward objectives as future work. Perhaps the reviewer could suggest other objectives that we could test as future work?

---

> ### Author Response · Authors · 2024-11-22
> **Addressing reviewer's recommendations**
>
> > Use the same measures on their data that the experimental results they are attempting to explain use
>
> We have used the canonical density metric used in experimental paper (Gauthier et al. 2019) as requested to update the figures in the paper. The new figures are Fig. 1, Sup. Fig.1, Sup. Fig. 2, Sup. Fig. 3.
>
> > Consider dropping the neural drift part of the results, as the underlaying mechanism in the model is at odds with many aspects of the hippocampal literature
>
> As mentioned in our response above, we respectfully disagree that place fields are generally stable over days and do not drift along the track. Furthermore, Reviewer Zibp mentioned that the “model is capturing the robustness against drift and modeling potential normative benefits of drift is a nice contribution -- this has been hypothesized in a number of places (eg Driscoll, Duncker et al 2022) and is well-demonstrated in the deep learning context (as the paper cites), but I don't know of any computational work demonstrating it in the context of a hippocampal navigation model” and, LQgu mentioned that “use of noise to modulate the learning flexibility is interesting and potentially insightful for neuroscience”. Hence, we believe the drift result is a key contribution.
>
> > Consider focussing on a set of experimental results better suited to the model e.g. the Dupret et al 2010 cheeseboard task
>
> Thank you for the suggestion. Fig. 1C is a similar analysis to Dupret et al 2010, where we observe place fields' shifting towards the reward location in a 2D arena to increase goal representation during learning.

---

> > ### Comment · Reviewer_ifiL · 2024-11-27
> > **Response to Authors**
> >
> > I appreciate the effort the effort the authors have put into addressing the concerns from the reviewers here, and I believe the method and approach here has promise to provide a useful model of hippocampal function. However, while I believe it could, in it's current form I do not believe the work provides a meaningful contribution to understanding representations in the hippocampus.
> >
> > I am happy to see the improvement to modelling place field density (measured as com) at goal locations, albeit I am a little troubled as to how and why the initial example and figures shown in the paper displayed the opposite of this.
> >
> > My main issue now is this - as explained in the author's rebuttal, they clearly understand how the model makes detailed (primary) predictions about the specific reorganisation of place fields based in their position in the rewarded linear track environment, and this is the mechanism behind the (secondary) effect they report that is the neural drift. Rather than comparing this secondary effect, I implore the authors to spend more time comparing their model to actual hippocampal data and testing the specific primary predictions their model makes. Providing a mechanistic explanation for the the drift of place fields on a track would be a valuable  contribution, but merely noting this has an effect on the population vector that is similar to neural drift is somewhat incomplete.

---

> ### Author Response · Authors · 2024-11-28
> **Clarification about density example**
>
> > I appreciate the effort the effort the authors have put into addressing the concerns from the reviewers here, and I believe the method and approach here has promise to provide a useful model of hippocampal function. However, while I believe it could, in it's current form I do not believe the work provides a meaningful contribution to understanding representations in the hippocampus.
>
> We thank the reviewer for the clarification and considering our initial response. We sincerely hope the next few points can help better clarify the contributions.
>
> > I am happy to see the improvement to modelling place field density (measured as com) at goal locations, albeit I am a little troubled as to how and why the initial example and figures shown in the paper displayed the opposite of this.
>
> We agree that the initial example was troubling as we had only considered the mean firing rate metric as indicative of density and did not consider the COM. We should have chosen an example where both density and mean firing rate were high at the reward location. We have replaced that example in Fig. 1B with another example that shows this.
>
> Furthermore, we describe in the revised manuscript that when the agent has a low number of fields (less than N<32) and are homogeneously distributed, a high density does emerge at the reward location early in the learning phase (Sup. Fig. 1A, T<5000). As learning proceeds, fields tend to shift backwards from the reward location to the start location (Sup. Fig. 1A, 5000<T<50000) so as to place more emphasis on the start location as credit assignment by the TD error occurs. Instead, when the number of fields is higher with a homogenous population (N>64, Sup. Fig. 1B), or when the place field distribution is heterogeneous and with a low number of fields (N>8), we robustly get a high density and mean firing rate at the reward location followed by the start location (Sup. Fig. 2).
>
> We thank you and other reviewers for highlighting this concern and we hope we have addressed this.

---

> ### Author Response · Authors · 2024-11-28
> **New model analysis to replicate hippocampal data**
>
> > My main issue now is this ... I implore the authors to spend more time comparing their model to actual hippocampal data and testing the specific primary predictions their model makes. Providing a mechanistic explanation for the the drift of place fields on a track would be a valuable contribution, but merely noting this has an effect on the population vector that is similar to neural drift is somewhat incomplete.
>
> We would like to clarify that the reorganization driven by reward maximization (the primary mechanism) alone is not sufficient to cause a significant decrease in the population vector correlation after stable navigation behavior is reached, i.e., after trial 25,000 in our simulations. Each place field's spatial selectivity remained fairly stable, as seen by the blue line in the PV correlation plot (Fig. 3B). Instead, we had to introduce Gaussian noise of varying magnitudes to each place field's center, width, and amplitude (Appendix D) as a primary mechanism to cause each field's spatial selectivity to change over time (Sup. Fig. 8G), consequently inducing a faster decrease in PV Corr. Nevertheless, we agree with the reviewer that injecting Gaussian noise to induce a faster decrease in PV Corr is insufficient to claim representative drift.
>
> To address your concern, we have conducted several new analyses of our model against actual hippocampal data and included these as the new Sup. Fig. 9, as requested. First, we show that the place fields in our model replicate the gradual change in spatial selectivity across days, as seen in Ziv et al. 2013 Nature Neuro. We pooled place fields from 10 agents (N=64 each) initialized with different seeds to track the dynamics of 640 fields (Sup. Fig. 9B), while Ziv et al. 2013 similarly pooled place fields across four mice (Sup. Fig. 9A). A small Gaussian noise of $\sigma_{\text{noise}}=0.00025$ was introduced to each place field parameter across learning. Similar to the neural data, place fields gradually changed their spatial selectivity as learning progressed, i.e., the spatial selectivity of cells (model's fields) sorted according to day 5 (trial 25,000) is more similar to day 10 (trial 75,000) than to day 35 (trial 195,000).
>
> Ziv et al. 2013, Geva et al. 2023 Neuron, and Qin et al. 2023 Nature Neuro, who analyzed data from Gonzalez et al. 2019 Science, further showed that each centroid gradually shifted away from its initialized position over a matter of days, indicating that fields do drift along the 1D track. This shifting dynamic is different from the null hypothesis of randomly shuffling field centroids (Sup. Fig. 9C). In our model, when no noise is added to field parameters $\sigma_{\text{noise}}=0.0$, we do not observe similar shift dynamics in centroid position (Sup. Fig. 9D). Instead, adding a small magnitude of noise $\sigma_{\text{noise}} \in \{0.0001, 0.00025, 0.0005\}$ replicates the shift dynamics observed in neural data. When a larger noise magnitude is used $\sigma_{\text{noise}} \geq 0.001$, the drift dynamics become similar to the null hypothesis of randomly shuffling centroids. Hence, we believe the model replicates the shifting dynamics observed in actual hippocampal data.
>
> Lastly, Gonzalez et al. 2019 Science demonstrated that although place fields changed their spatial selectivity over time, the ensemble representation was stable by maintaining clusters to encode spatially relevant information (Sup. Fig. 9E). Similarly, we visualized the topology of our model's place field centroids, which were initialized along the 1D track to encode the reward (red nodes) and start (blue) locations. As learning progressed, the ensemble representation rotated, and the cosine distance between individual centroids changed. Nevertheless, reward and start location coding fields maintained their clustering, which became more separated as learning progressed, further supporting the emergence of a high density of fields at the reward and start locations (Sup. Fig. 9F).

---

> ### Author Response · Authors · 2024-11-28
> **Additional discussion about manuscript's contribution**
>
> Furthermore, based on reviewer ZibP's request, the model shows that a small proportion of fields coding for an initial reward location shifts to encode the new reward location while a high density of fields emerges at the new reward location, similar to the hippocampal data seen in Gauthier et al. 2018 Neuron (newly included as Sup. Fig. 3).
>
> Besides these additional simulations, we have included a discussion about the biological origins of adding Gaussian noise to place field parameters as requested by other reviewers. We observe that synaptic plasticity is noisy due to several molecular mechanisms and pathways (Allen \& Stevens 1994; Mongillo, Rumpel, Loewenstein 2017; Kappel, Habenschuss, Legenstein, Maass 2015; Attardo, Fitzgerald, Schnitzer 2015). Additionally, extrinsic factors such as glia (Sancho, Contreras, Allen 2001) and astrocytes (Bernardinelli, Muller, Nikonenko 2014; Pitta, Brunel, Volterra 2016) also modulate synaptic plasticity. These experimental data suggest that it is reasonable to add small Gaussian noise to synapses or neural parameters during learning.
>
> Lastly, we would like to clarify that our initial focus was to determine if there could be a functional role for fields changing spatial selectivity, as similarly observed in a non-hippocampal context by Dohare et al. 2024 Nature. Specifically, would having dynamically drifting place fields endow the agent with improved navigation performance? Our simulations (Fig. 4C, Sup. Fig. 11, Sup. Fig. 12) showed that increasing the magnitude of Gaussian noise increased the rate of decrease in population vector correlation. However, we observed a non-monotonic improvement in the agent's flexibility in learning new targets in both the 1D and 2D arenas. Nevertheless, we agree with the reviewer that we should have conducted more analysis to ascertain that our model captures the drift dynamics observed in hippocampal data, before answering this question.
>
> We sincerely thank the reviewer once again for their critical feedback. We trust that we have addressed the main concerns to a reasonable extent while being mindful of the timeline constraints. If there is a specific paper on drift dynamics that you consider particularly compelling in the context of neural drift, please do let us know. We kindly request that you reconsider increasing our submission's score. Rest assured, we will continue analyzing the model to further align its reorganization dynamics with observed neural data.

---

### Official Review · Reviewer_Zibp · 2024-11-09

**Soundness:** 3
**Presentation:** 3
**Contribution:** 3
**Rating:** 5
**Confidence:** 4

**Summary:**

This paper uses a representation learning perspective to understand how the reorganization of place fields under experience and reinforcement supports RL. The authors aim to recapitulate three experimentally observed phenomena: the timeline of dense place fields emerging at rewarded locations, lengthening of fields against the direction of motion, and stable performance amidst place field drift. They also include a number of additional experiments and analyses characterizing features of the model.

**Strengths:**

- Overall it's a nice approach to relating hippocampal activity to downstream RL computations. Also seems like a very plausible constraint on neural activity. The model is minimally complex while still allowing insight on representation learning
- Model is capturing some phenomena in the data very nicely.
- In particular, the model is capturing the robustness against drift and modeling potential normative benefits of drift is a nice contribution -- this has been hypothesized in a number of places (eg Driscoll, Duncker et al 2022) and is well-demonstrated in the deep learning context (as the paper cites), but I don't know of any computational work demonstrating it in the context of a hippocampal navigation model.

**Weaknesses:**

Weaknesses

- unclear comparisons / mismatches with data.

First, the density of the place fields that is reported typically measures the number of fields with centers near the goal, not the total summed firing at a location. The prediction of increased place field density at the reward location should therefore be a larger number of place fields with centers near or before the goal. The plots in Fig 1 seem to actually show the opposite (with the possible exception of T=1000) -- place fields *spread out* near the goal location but get much larger.

Also, Gauthier & Tank report that these cells are reward specialized, and therefore remap with reward. I believe that would not be a prediction of this model -- enabling this would require adding reward / reward-predictive landmarks as as an observation dimension and observe this.

Authors note that there is a rapid increase in the density of place fields near the goal location which quickly changes, with fields spreading out a lot and getting larger amplitude. I don't believe the latter observations are consistent with the data.

It appears that the number of place fields associated with non-rewarding, but frequently visited regions might go down. I believe this is not consistent with data, where home locations (and even salient landmarks) have many place fields.

- robustness

Does the same thing happen each time? no error bars in a lot of the plots (particularly regarding density).

- confusing benchmarks

Only one benchmark -- SR fields. This was described as predicting future location occupancy p(x_{t+1} | x_t), but I believe the SR is in fact encoding a discounted sum of future occupancy \sum_{t=0}^\infty \gamma^t p(x_{t+1}|x_t) (aka, the expected number of visits to future states).

Another relevant benchmark is also Fang et al 2024 which also do representation learning in a deep RL framework, and show clustering at reward and skewing (but do not look at drift / skew against the direction of motion).

- needs tidying in some places

eperience
hippocamppal
weighs -> weights
delta defined indirectly in expectation
"The agent’s transition in the environment is smooth as we use a low-pass filter using a constant" didn't understand this.

- argument about number of fields seems speculative. Authors argue it could be because it's in weak feature learning regime, but don't see any evidence of that (and an efficient coding hypothesis seems more likely).

- T's not matched up for Fig 2C?

Comments.

- Consider adding dropout rather than / in addition to parameter drift, as i think that will lead to more distributed representations and more actual place field clustering near goal (and also consistent with neural activity, see low, lewallen et al 2018).

**Questions:**

- What is going on with the giant place field in Fig 2C?

> "To determine an agent’s reward maximization performance during navigational learning we track the true cumulative discounted reward"

why not just reward rate?

- Place fields form in the absence of any reward, and many characteristics of place fields with respect to reward appear to also occur for salient landmarks. Time spent in a region of space and sensory complexity also alters place fields in the absence of reward. What happens to place fields in this model absent reward?

---

> ### Author Response · Authors · 2024-11-22
> **Model reproduces high density (number) of place fields at reward location**
>
> We would like to thank the reviewer for their insightful comments. We have rectified Fig. 1-2, Sup. Fig. 1,5 8, added new Sup. Fig. 2-4, and uploaded the new manuscript with the recommended references based on your comments, which we will refer to henceforth.
>
> > First, the density of the place fields that is reported typically measures the number of fields with centers near the goal, not the total summed firing at a location. The prediction of increased place field density at the reward location should therefore be a larger number of place fields with centers near or before the goal. The plots in Fig 1 seem to actually show the opposite (with the possible exception of T=1000) -- place fields spread out near the goal location but get much larger.
>
> > Authors note that there is a rapid increase in the density of place fields near the goal location which quickly changes, with fields spreading out a lot and getting larger amplitude. I don't believe the latter observations are consistent with the data.
>
> Yes, density is described as the number of fields in a location (Gauthier et al. 2018; Lee et al. 2020). We defined density as the mean firing activity at a location instead of the number of fields (based on the center of mass) as a higher firing rate (amplitude) at a location influences policy learning to a greater extent (Sup Fig. 8C) than a place field’s center of mass or a single field’s firing rate alone. That is, a small number of place fields with large firing rates is more influential than a large number of fields with extremely small firing rates. Furthermore, optimizing for a field’s center of mass is not as beneficial in policy learning as compared to a field’s width and amplitude. Instead, optimizing for a field’s center destabilizes policy learning (Fig. 4A). Hence, we wanted to motivate a more holistic definition of density for policy learning.
>
> Nevertheless, we agree that we should have used the canonical density definition (number of fields in a location) instead. Fig. 1 and Sup. Fig. 1 in the paper have been re-plotted with the density defined as the number of fields with their center of mass in a location as in the literature. The blue line in the new figures show the number of place fields in a location measured using their Center of Mass (COM) and smoothed using a Gaussian kernel density estimate d(x) = gKDE(COM). Based on the canonical density definition, we do concede that the example distribution shown in the original Fig. 1B shows a lower density at the reward location, as pointed out. However, we would like to argue that this is not a good example and instead, a high density of fields do in fact emerge at the reward location robustly as we observe in data.
>
> In the newly attached Fig. 1B,C and Sup. Fig. 1, we show how optimizing different combinations of place field parameters (center $\lambda$, amplitude $\alpha$,width $\sigma$) influences place field reorganization. When we optimize a low number of fields e.g. N=16 that have been uniformly distributed along the 1D track with the same width and amplitude, a higher number of fields with small firing rates organize at the home location (Sup. Fig. 1A), contrary to neural data. However, when we increase the number of tunable fields that are uniformly distributed along the 1D track with the same width and amplitude e.g. N=256, a high number of fields do organize at the reward location (Sup. Fig.1B), similar to neural data.
>
> More importantly, we randomly sampled each place field’s amplitude, center and width parameters from a uniform distribution between [0,1], [-1,1] and [1e-5, 0.1] respectively and optimized the parameters using the same algorithm over 50 different seeds. The new Fig. 1B and Sup. Fig. 2A,B and. shows that a high density (number) of fields do emerge at the reward location followed by the home location robustly across the 50 seeds when we optimize 4 to 512 place fields.

---

> ### Author Response · Authors · 2024-11-22
> **Theory predicts fields move to the reward and start location**
>
> Additionally, we would like to refer back to Eq. 7 and Appendix B that derives a perturbative approximation of our place field model. A theoretical analysis of the gradient flow shows that place field centers will straddle between a frequently visited location \bar \lambda i.e. reward location and the start location \mu_x, depending on its relative position in the track.
>
> Fields will shift depending on the value of the location (indicated by the magnitude in $w_{v,i}$). In the beginning of learning, the reward location will have a non-zero, positive value while the home location tends to have a value close to zero. Hence, fields near the reward location will move first. In the later stages of learning, when the home location increases in value, due to credit assignment by the TD error, fields near the home location will shift.
>
> The direction of shift depends on the location in which the field was initialized at. When a field is initialized near the reward location ($\bar \lambda ~ \lambda$ ) , it will have a tendency to move towards the reward so that the difference $\bar \lambda - \lambda$ goes towards zero causing $\Delta \lambda(t)$ in Eq. 66 to go towards zero. However, $\mu_x - \lambda$ will be nonzero, causing a pull on that field to move closer to the home location $\mu_x$. Similarly, when a field is initialized near the home location ($\mu_x ~ \lambda$), it will have the tendency to move towards the home location while having a pull towards the reward location. Hence, each field will continue to dynamically change its location until a stable distribution is achieved that straddles between the reward location and home location. Based on simulations, when there is a low number of fields, the gradient optimization process tries to balance field centers between the reward and start location, though an equilibrium point might not be achievable. When the number of fields is high, an equilibrium point can be attained.
>
> This equation also explains why we see a backward shift in the field center towards the home location in later stages of learning. We hope that using the canonical density metric, new simulations and theoretical analysis in the paper will be sufficient to argue that our model does show a higher number of fields organizing both at the reward and home locations as seen in neural data. We look forward to the reviewer’s perspective on these.

---

> ### Author Response · Authors · 2024-11-22
> **A small number of reward-encoding fields do shift to the new reward location.**
>
> > Also, Gauthier & Tank report that these cells are reward specialized, and therefore remap with reward. I believe that would not be a prediction of this model -- enabling this would require adding reward / reward-predictive landmarks as as an observation dimension and observe this.
>
> Yes, that is correct. We would need additional mechanisms to anchor the place fields to landmarks, which the current task and model lacks.
>
> Interestingly, in our current model, a small proportion of fields (2.6%) initially encoding the reward location at $x_r=0.75$ shifted to the new reward location at $x_r=-0.2$ (Sup. Fig. 3B). Furthermore, a high density (number) of fields emerged at the new location after the shift (Sup. Fig. 3A). There was a sizable number of fields that  did shift backwards, though not completely to the new reward location. If learning was to continue, perhaps a higher proportion of place fields encoding the initial reward will shift backward to the new reward location. The current proportion might not be as high as observed in the experimental data (Fig. 6E in Sosa et al. 2023 bioRxiv). However, we believe our model might still be useful to understand the mechanism of this shift. We have added these figures to the supplementary figures and in the discussion.
>
> Additionally, we have an idea to model place fields with conjunctive spatial and non-spatial selectivity. This architecture was previously explored by Kumar et al. 2022 Cerebral Cortex whereby CA3 place cell activity and non-spatial/sensory Entorhinal cortical activity were passed to either a feedforward or recurrent network with an actor-critic output. The feedforward units had conjunctive selectivity and had place field-like activity. However, the representation was not optimized but was kept fixed. We are excited to evaluate reward coding and shift dynamics using this architecture as a follow up.

---

> ### Author Response · Authors · 2024-11-22
> **Field density does increase at home location**
>
> > It appears that the number of place fields associated with non-rewarding, but frequently visited regions might go down. I believe this is not consistent with data, where home locations (and even salient landmarks) have many place fields.
>
> The newly attached Fig. 1, Sup. Fig. 1 and Sup. Fig. 2 shows that the number of place fields associated with the home locations increases as learning occurs for both low and high number of fields, consistent with data (Gauthier et al. 2018). Optimizing different place field parameter combinations shows a consistent increase in the number of fields at the home location, with the reward location having the highest density, followed by the home location. Furthermore, our perturbative approximation shows that place fields will move to the home location $\mu_x$ and to locations that are frequently visited i.e. reward location $\lambda_x$. Hence, the number of fields associated with non-rewarding, home location increases, albeit at a slower rate than the rewarding region, consistent with data. May we know how else we can best clarify this result?

---

> ### Author Response · Authors · 2024-11-22
> **Robustness of the model**
>
> > Does the same thing happen each time? no error bars in a lot of the plots (particularly regarding density).
>
> Yes, in Sup. Fig.2, we see the same high density (number) emerging at both reward location and non-reward home location across 50 different seed iterations and place fields with amplitude, center and widths initialized using a random uniform distribution between [0,1], [-1,1] and [1e-5, 0.1] respectively.
>
> Fig. 1B and C were examples of a single run while Fig. 1D shows how the density changes over 50 different seeds. We have added error bars for Fig. 1D and shaded area for Fig. 1B-C and Fig. 2C. We had error bars and shaded area to indicate Confidence Interval of 95% for Fig.3, Fig.4 and supplementary figures. Our theoretical analysis of the perturbative approximation of the model further adds to the robustness of the claim. Would the theory and empirical simulations with error bars be sufficient?

---

> ### Author Response · Authors · 2024-11-22
> **Reward maximization objective reproduces dynamics observed in non-reward based model**
>
> > Only one benchmark -- SR fields. This was described as predicting future location occupancy p(x_{t+1} | x_t), but I believe the SR is in fact encoding a discounted sum of future occupancy \sum_{t=0}^\infty \gamma^t p(x_{t+1}|x_t) (aka, the expected number of visits to future states). Another relevant benchmark is also Fang et al 2024 which also do representation learning in a deep RL framework, and show clustering at reward and skewing (but do not look at drift / skew against the direction of motion).
>
> The SR has been used as the gold standard/baseline of a biologically plausible predictive mechanism to describe the elongation of fields against trajectory (Stachenfeld et al. 2017 Nature. Neuro.). There are other learning rules e.g. STDP (Mehta et al. 2000), BTSP (Madar et al. 2023) that describe the elongation of fields against trajectory. However, we felt it would be appealing to compare our model against the well-known SR algorithm, as also supported by Reviewers ifiL, J18N and LQgu.
>
> Importantly, we wanted to demonstrate that optimizing for a reward maximizing objective could replicate the dynamics observed when using a non-reward based mechanistic model i.e. SR, which is counter-intuitive and surprising. That perhaps the hippocampus might be optimizing for an intrinsically defined reward metric, hence “Reward Maximization” in the title.
>
> Nevertheless, you are correct that we should definitely explore non-reward maximizing normative objectives that the hippocampus could be optimizing for such as the ones used in Fang et al 2024 ICLR and Foster et al. 2000 Hippocampus. We are excited to explore these non-rewarding objectives in a follow-up paper, and we have included this suggestion as a future direction in the discussion section.

---

> ### Author Response · Authors · 2024-11-22
> **Clarifications on agent dynamics and weak feature learning**
>
> > eperience hippocamppal weighs -> weights delta defined indirectly in expectation
>
> Thank you for pointing out these mistakes. We have rectified these errors in the updated manuscript.
>
> > "The agent’s transition in the environment is smooth as we use a low-pass filter using a constant" didn't understand this.
>
> The agent outputs a one-hot vector g_t to indicate which direction (left or right) the agent should move. This causes the agent’s motion to be discrete, similar to a trajectory in a grid world. To model smooth trajectories in a continuous space as in animal behavior, we smoothened the agent’s one-hot output g_t using a low-pass filter $a_{t+1} = (1-\alpha_{env}) *a_t + alpha_{env} * g_t $ to get a vector $a_t$ so that the transition x_{t+1} = x_t + a_t is smooth instead of discrete, similar to Foster et al. 2000 Hippocampus and Kumar et al. 2022 Cerebral Cortex. We have re-written this part in the updated manuscript.
>
> > argument about number of fields seems speculative. Authors argue it could be because it's in weak feature learning regime, but don't see any evidence of that (and an efficient coding hypothesis seems more likely).
>
> We consider a weak place field feature learning regime when each place field’s parameter \theta \in \{\alpha, \sigma, \lambda\} does not change from their initialized values. Sup. Fig. 4 shows that as the number of place fields increases, the change in Euclidean distance or L2 distance for each parameter decreases, suggesting that that when the number of place fields are high, they do not show a significant change in their parameters and hence being in a weak feature learning regime. $||\theta_{final} - \theta_0|| = ||\alpha_{final} - \alpha_{init} || + ||\lambda_{final} - \lambda_{init} || + ||\sigma_{final} - \sigma_{init} || $. We have included this figure as a Sup. Fig. 4 in the paper.
>
> > T's not matched up for Fig 2C?
>
>  Due to space constraints, and to compare the dynamics of field distribution during learning, we chose T= 10,000 for SR (tor row) and T=3000 for RM (bottom row) as the intermediate distribution. For the SR model, T=3000 (Top row) showed a similar distribution to T=1000. For the RM model, T=10,000 (bottom row) was similar to T=50,000. We have included the figures for T=100,1000, 3000, 10000, 50000 as Sup. Fig. 5E for both SR and RM for clarity.

---

> ### Author Response · Authors · 2024-11-22
> **Clarification about modeling representational drift using dropout.**
>
> > Consider adding dropout rather than / in addition to parameter drift, as i think that will lead to more distributed representations and more actual place field clustering near goal (and also consistent with neural activity, see low, lewallen et al 2018).
>
> Thank you for the excellent data reference. We have added this reference to the discussion section that requires further analysis. May we clarify how we should add dropout to the model i.e. should we add dropout during place field parameter update, randomly silence a random subset of place fields so that individual place fields become silent at different timesteps, or silence the synapses from place fields to the actor and critic network as in deep networks?
>
> In this paper, we sampled Gaussian noise with mean=0 and various variance magnitudes and added them to the place field amplitude, width and amplitude parameters during the update stage (Appendix 3) similar to Qin et al. 2023 Nature. Neuro. to induce representational drift. By doing so, our place field model captures 1) individual place fields moving away from their initialized position (Sup. Fig. 8G left, change in $\lambda_i$), 2) each place field’s firing rate fluctuates such that they lose or gain the place field (Sup. Fig. 8G center, change in $\alpha_i^2$), and 3) individual field’s size fluctuates while most of them increases with learning (Sup. Fig. 8G right, change in $\sigma_i^2$). (1) and (2) has been observed in neural data as shown by Qin et al. 2023. Could this be an analogue to the effects of dropout?
>
> Based on this clarification, we can try to do a quick experiment or add this as a suggestion to the discussion section and explore this method as a future study.

---

> ### Author Response · Authors · 2024-11-22
> **Giant place field was expected to group locations with the same actions into a single state**
>
> > What is going on with the giant place field in Fig 2C?
>
> As seen in Eliav et al. 2021 Science, place fields have heterogeneous field sizes with the largest field spanning up to 32 meters and the smallest spanning 0.6 meters. To determine if place fields that were initialized uniformly could learn to become heterogeneous purely using gradients, we did not impose any bounding limits or constraints to the place field parameters in the 1D track. This led to the growth of large place fields starting from the reward and extending backwards towards the home location.
>
> We would also like to highlight that large place fields are expected from a state representation learning point of view where these fields are grouping locations that require the same action into a singular state to improve the efficiency of policy learning i.e. we only need 2 place fields to solve the 1D navigation task, one to move right from the home location to the reward location and another to stop at the reward location. Hence, we also consider these place fields to be performing a form of state representation learning. Importantly, field elongation leads to the biggest improvement in policy learning as seen in Fig. 4.
>
> Additionally, when analyzing place fields in a small environment, we often discard cells that are active throughout the track from analysis. Based on our model’s predictions that large place fields will emerge, we will be analyzing the dynamics of all the cells that have been recorded to determine if a similar elongation throughout the 1D track occurs.
>
> > "To determine an agent’s reward maximization performance during navigational learning we track the true cumulative discounted reward" why not just reward rate?
>
> We used cumulative discounted reward as this was the same term/objective that the agent was maximizing for during policy learning and place field optimization $J = E[\sum_t \gamma^k r_{t+1+k}]$. This was to demonstrate that the agent is indeed maximizing the objective. Tracking just the reward rate shows the same learning performance as tracking the reward rate. We can change the performance metric to the reward rate for clarity.

---

> ### Author Response · Authors · 2024-11-22
> **Optimizing place fields using non-reward based objectives**
>
> > Place fields form in the absence of any reward, and many characteristics of place fields with respect to reward appear to also occur for salient landmarks. Time spent in a region of space and sensory complexity also alters place fields in the absence of reward. What happens to place fields in this model absent reward?
>
> We agree that place fields do reorganize in the absence of rewards, as a form of latent learning (Tolman, 1948). However, in our current model, when rewards are absent, place fields will not show much reorganization since the objective that the place fields are optimizing for are reward dependent. We started with a reward based objective as most theory in reinforcement learning is centered around rewards (Sutton & Barto 2018; Kakade 2023; Agarwal et al. 2021; Bordelon et al. 2023 NeurIPS), and we were curious if a reward-like learning signal could replicate disparate neural phenomena, even those that are thought to not require rewards.
>
> Nevertheless, we believe the current model can be easily extended to non-reward based objectives since we perform gradient computation, and theoretically study (E.g. Eq. 7 and Eq. 65) how these objectives influence place field reorganization in a follow up paper. As a starting point, we will explore the objectives used in Fang 2024 ICLR which is to predict the next state given the current state, and Foster et al. 2000 Hippocampus which is to predict a low-dimensional metric representation from place field activity. We have included our model’s reward dependency as a limitation in the discussion section and have described exploring non-reward based objectives as future work.

---

> > ### Comment · Reviewer_Zibp · 2024-11-27
> > **Response to authors**
> >
> > The authors new comments thoroughly addressed a number of my comments, and I have incremented my score from 3->5. Thank you for your hard work on this rebuttal. However, I do still have a number of outstanding concerns:
> >
> > 1. my biggest remaining concern regards the implementation of the SR benchmark. To clarify, I don't have any issue with the use of the SR as a benchmark here. My concern was actually that I think this equation is just not the correct implementation of the SR, which is the summed future occupancy, rather than the expected next state transition matrix.
> >
> > I believe the equation is this: \sum_{t=0}^\infty \gamma^t p(x_{t+1}|x_t).
> >
> > I agree it's not necessary to implement the model from Fang et al 2024, especially as it is recent work, but it is relevant in that it also explains similar place field effects with representation learning under deep RL perspective and bears mention.
> >
> > 2. my second concern regards the interpretation. it seems the SR and the RM model make very similar predictions about the distribution of fields and their dynamics on aspects that have been experimentally verified. Where the models depart is on the dynamics of when a high density of cells emerges at different places. Here, it is not clear which one better resembles the data.
> >
> > It makes sense that this model will not capture all place field effects, in that it is scoped to capture what reward maximization contributes to place field reorganization. But it would help to have clarity about what data this model uniquely captures (or if it's rather intended to provide an alternate account).
> >
> > > Density
> >
> > The new experiments and new metric address my concerns here. You can ignore the dropout suggestion -- it was just a thought about how to encourage distributed representations. The constraints implemented in these experiments suffices though.
> >
> > > reward-specialized cells remapping with reward
> >
> > Response here is fine -- will want to mention this discrepancy in the paper though if it's not already.
> >
> > > Authors note that there is a rapid increase in the density of place fields near the goal location which quickly changes, with fields spreading out a lot and getting larger amplitude. I don't believe the latter observations are consistent with the data.
> >
> > The analyses on the dynamics of the field dynamics are super interesting here, and are a nice addition to the paper. I still want to know if this is consistent with observed data about the dynamics of place field emergence.
> >
> > > It appears that the number of place fields associated with non-rewarding, but frequently visited regions might go down. I believe this is not consistent with data, where home locations (and even salient landmarks) have many place fields.
> >
> > addressed
> >
> > > robustness
> >
> > addressed
> >
> > > What is going on with the giant place field in Fig 2C?
> >
> > Yes, place fields are heterogenous in size. However, the observed distribution of place field sizes here is not the observed distribution of place field sizes, which I believe are log-normal and exhibit a hyperbolic geometry. This paper also notes that place field density expands with time spent at a location (rather than reward).
> > https://www.nature.com/articles/s41593-022-01212-4
> >
> >
> > > Place fields form in the absence of any reward, and many characteristics of place fields with respect to reward appear to also occur for salient landmarks. Time spent in a region of space and sensory complexity also alters place fields in the absence of reward. What happens to place fields in this model absent reward?
> >
> > Addressed

---

> ### Author Response · Authors · 2024-11-27
> **Data analysis to verify model's predictions**
>
> We would like to thank the reviewer once again for insightful feedback and for raising the score! The manuscript is in a much better form.
>
> With regards to SR, we agree that we have not performed data analysis to determine which model fits the data better. Hence, in the current state, this serves as a prediction to verify. We will compare the representations learned by SR, RM and Fang et al. 2024 models in the follow up paper which will include data analysis.
>
> We have mentioned the discrepancy in place fields reward mapping behavior between the model and the data in the manuscript.
>
> With regards to drift dynamics, this is yet another prediction of the model that we are eager to analyze in the follow up paper.
>
> We do see that only a small proportion of fields expand to become a giant place field. We will analyze the model behavior to determine if the place field expansion follows the log-normal distribution. Thank you for the reference.
>
> Please feel free to let us know if you have any other thoughts to improve the manuscript.

---

### Author Response · Authors · 2024-11-26
**Summary of revisions**

We would like to thank all the reviewers and area chairs for taking the time and effort in reviewing our manuscript and providing insightful feedback. We have clarified each reviewer's comments and have revised the manuscript with new experiments, figures, analyses and discussions. Below, we give an overview of the common and specific questions by each reviewer. If you have any additional comments or questions, please let us know so that we can address them before the deadline for revisions this Wednesday. We hope the modifications will be sufficient to reconsider the score.
Kindly, Submission 10547

### Common concerns:
- Use the definition of density in the literature (Reviewers Zibp, ifiL):
-- Modified Figs 1B, 1C, 1D, Sup. Fig. 1
-- New Sup. Fig. 2
-- Modified text 187-205 and Figure 1 captions

- Modeling drift dynamics observed in the literature (Reviewers Zibp, ifiL):
-- New Sup. Fig. 8G

- Biological origins of noise (Reviewers J18N, LQgu):
-- Added discussion in lines 510-513

- Feature learning regime (Reviewers Zibp, J18N):
-- New Sup. Fig, 4

- Consideration of non-reward based objective for place field reorganization (Reviewers Zibp, ifiL, LQgu):
-- Added discussion in lines 529-531

### Specific questions:
- Whether place fields remap when reward location changes (Reviewer Zibp):
-- New Sup. Fig, 3

- Clarifying agent transition and field dynamcs (Reviewer Zibp):
-- Clarified lines 111-120
-- New Sup. Fig. 5E

- Whether using the same learning rate will change phenomena (Reviewer J18N):
--New Sup. Fig. 12

- Other models of place fields e.g. center-surround (Reviewer J18N):
-- New Sup. Fig. 13

- Biological plausibility of model (Reviewer LQgu):
-- New Sup. Fig. 14.

- Code for agents (Reviewer LQgu):
-- We have included a minimal working code as a .zip folder to train agents in the 1D track and 2D arena with obstacle.

---

### Author Response · Authors · 2024-12-03
**Additional Analyses**

We sincerely appreciate all the reviewers' thorough evaluation and insightful feedback.

In response to Reviewer ifiL's recommendations, we have conducted additional analyses to further validate our model's ability to replicate the drift dynamics observed in neural experiments. The results of this analysis are presented in Supplementary Figure 9 based on the findings from Ziv et al. (2013, Nature Neuroscience), de Snoo et al. (2023, Current Biology), Qin et al. (2023, Nature Neuroscience), and Gonzalez et al. (2019, Science).

We trust that this additional analyses addresses the reviewer's concerns and respectfully ask for reconsideration of the score.

---

### Meta-Review · Area_Chair_67wu · 2024-12-24

**Metareview:**

This paper presents a reinforcement learning model investigating how place fields, represented as Gaussian radial basis functions, reorganize through TD errors during navigation. The model aims to reproduce three key hippocampal phenomena: 1) increased place field density near rewards, 2) field elongation opposite to movement direction, and 3) stable place-field drift. Following author revisions, the model successfully demonstrated a higher number of place fields emerging at both reward and home locations, backed by theoretical derivations showing how fields dynamically balance between these locations. The authors also provided extensive robustness analyses across multiple experimental conditions and parameter settings.

However, several concerns led to the paper's rejection despite the authors' thorough revisions. While the authors addressed issues regarding place-field density measurements and home location representation, key limitations remained: the model's strict reliance on reward-based learning (unlike biological systems where place fields form without rewards), incomplete validation of drift dynamics against experimental data, and discrepancies in how reward-specialized cells remap compared to empirical observations. The reviewers noted that while both the successor representation (SR) and reward maximization (RM) models made similar predictions about field distribution and dynamics, additional data analysis would be needed to determine which better matches experimental observations. They recommended strengthening empirical validation and exploring non-reward learning mechanisms before resubmission.

**Additional Comments On Reviewer Discussion:**

Throughout the rebuttal, the authors made significant revisions to better align their work with empirical evidence and address reviewer concerns. A key change was redefining place-field density to use center-of-mass counts instead of summed firing rates, matching established experimental measures. They supplemented this with new plots showing how moderate noise injection could reproduce gradual drift patterns, supporting this with references to studies demonstrating place-field centroid shifts over days. Their additional analyses strengthened the argument that reward-focused objectives can drive field emergence near goal locations, though they acknowledged the limitation that hippocampal fields also reorganize in non-reward contexts. These revisions specifically targeted reviewer concerns about biological alignment, drift modeling robustness, and the scope of reward-based learning.

However, reviewers were not convinced about how accurately the model captured real-world place-field density patterns near rewards, questioned the biological plausibility of the noise-driven drift mechanism, and expressed doubts about whether a purely reward-based approach could adequately explain hippocampal phenomena in non-reward scenarios. While the authors' supplementary analyses provided valuable clarification regarding noise magnitudes and place-field shifts, their efforts did not fully resolve concerns about empirical grounding and the model's breadth of applicability.

---

### Decision · Program_Chairs · 2025-01-22

Reject